# A new class of disordered elements controls DNA replication through initiator self-assembly

Matthew W Parker[1,2], Maren Bell[2], Mustafa Mir[2], Jonchee A Kao[2], Xavier Darzacq[2], Michael R Botchan[2]*, James M Berger[1]*

[1]Department of Biophysics and Biophysical Chemistry, Johns Hopkins School of Medicine, Baltimore, United States; [2]Department of Molecular and Cell Biology, University of California, Berkeley, Berkeley, United States

**Abstract** The initiation of DNA replication in metazoans occurs at thousands of chromosomal sites known as origins. At each origin, the Origin Recognition Complex (ORC), Cdc6, and Cdt1 co-assemble to load the Mcm2-7 replicative helicase onto chromatin. Current replication models envisage a linear arrangement of isolated origins functioning autonomously; the extent of inter-origin organization and communication is unknown. Here, we report that the replication initiation machinery of *D. melanogaster* unexpectedly undergoes liquid-liquid phase separation (LLPS) upon binding DNA in vitro. We find that ORC, Cdc6, and Cdt1 contain intrinsically disordered regions (IDRs) that drive LLPS and constitute a new class of phase separating elements. Initiator IDRs are shown to regulate multiple functions, including chromosome recruitment, initiator-specific co-assembly, and Mcm2-7 loading. These data help explain how CDK activity controls replication initiation and suggest that replication programs are subject to higher-order levels of inter-origin organization.

DOI: https://doi.org/10.7554/eLife.48562.001

*For correspondence:
mbotchan@berkeley.edu (MRB);
jberge29@jhmi.edu (JMB)

## Introduction

The appropriate spatiotemporal regulation of DNA replication is essential to genetic integrity and cell proliferation. In eukaryotes, the initiation of DNA replication requires the coordinated action of three proteinaceous factors – the Origin Recognition Complex (ORC), Cdc6, and Cdt1 – which co-assemble on DNA origins in the G1 phase of the cell cycle to catalyze loading of the Mcm2-7 replicative helicase onto chromatin. Once activated during the transition to S phase, Mcm2-7 helps promote origin melting, replisome assembly, and translocation of the replication fork.

ORC consists of a heterohexameric complex comprising the subunits Orc1-6. Five ORC subunits (Orc1-5), as well as Cdc6 and the six subunits of Mcm2-7, possess an ATPases Associated with diverse cellular Activities (AAA+) domain; Orc6 and Cdt1 are the only non-AAA+ proteins used for Mcm2-7 loading. Despite sharing a high overall degree of conservation across eukaryotes, certain aspects of initiator subunit sequence and function have nonetheless diversified during evolution. For example, *S. pombe* Orc4 contains a unique domain not found in other ORCs that endows the protein with a preference for A/T rich regions of DNA (*Chuang and Kelly, 1999*; *Lee et al., 2001*). Conversely, *S. cerevisiae* ORC is able to recognize specific origin sequences (*Bell and Stillman, 1992*; *Li et al., 2018*), whereas origin specification for metazoan ORCs appears more contextual (*Remus et al., 2004*; *Vashee et al., 2003*). *S. cerevisiae* Cdt1 additionally possesses a catalytically-inactive dioxygenase domain at its N-terminus that is necessary for yeast viability and Mcm2-7 loading (*Frigola et al., 2017*; *Takara and Bell, 2011*); this fold is absent in *S. pombe* and metazoan Cdt1s.

Much of our current understanding of initiator mechanism derives from reconstitution studies using budding yeast initiation factors. These efforts have helped define an orchestrated set of ATP-dependent molecular exchanges between ORC, Cdc6, and Cdt1 that culminate in the loading of two copies of Mcm2-7 onto origin DNA in the form of a stable double-hexamer (*Duzdevich et al., 2015*; *Evrin et al., 2009*; *Remus et al., 2009*; *Ticau et al., 2015*). One aspect of initiation that has been difficult to probe in vitro, however, is the potential for interactions between initiation factors associated with different origins. Evidence for mesoscale coordination of origin activity derives from multiple sources. For example, genomic studies aimed at defining sites of ORC binding and their relation to replication origins have revealed evidence of origin clustering across chromosomes at nucleosome-free regions (*Cayrou et al., 2011*; *Miotto et al., 2016*; *Vaughn et al., 1990*). Cellular patterns of ORC localization are also strikingly non-uniform, often presenting as concentrated foci on chromatin whether it be in *D. melanogaster* follicle cells (*Austin et al., 1999*) or human cells in tissue culture (*Lidonnici et al., 2004*; *Prasanth et al., 2010*; *Shen et al., 2010*). Outside of initiation, there exists a well-characterized phenomenon of origin interference that has been taken as evidence of inter-origin communication (*Cayrou et al., 2011*); the clustering of co-replicating regions into so-called Replication Factories or replication domains (RD) similarly has been observed in both fixed and live cells (*Cook, 1999*; *Xiang et al., 2018*). Finally, the replication timing profile of origins, which fire asynchronously throughout S-phase, partition to specific chromosome territories (*Cremer and Cremer, 2001*; *Gilbert et al., 2005*; *Pope et al., 2014*). It is currently unclear whether the clustering patterns observed for ORC binding sites, origin communication, and the timing of replication domains is coincidence or dependent upon an as-yet-discovered set of physiological properties of replication factors.

Present-day views of mesoscale organization within cells typically invoke processes such as protein gradients or membrane compartmentalization. Recently, a rapidly expanding body of work has begun to recognize protein/protein and protein/RNA liquid–liquid phase separation (LLPS) as playing critical functions in generating membraneless pseudo-organelles and co-localized bodies (*Boeynaems et al., 2018*). Both nuclear and cytoplasmic liquid phase condensates (also known as biomolecular condensates) have been observed and implicated in a panoply of functions, including cellular signaling (*Li et al., 2012*; *Su et al., 2016*), centrosome assembly (*Woodruff et al., 2017*), and chromatin/heterochromatin assembly and maintenance (*Larson et al., 2017*; *Strom et al., 2017*; reviewed in *Maeshima et al., 2016*). A functional role for biological condensates is only beginning to emerge; such entities may help sequester and/or co-localize certain factors to modulate biochemical output and response (*Shin and Brangwynne, 2017*). Intrinsically disordered amino acid regions (IDRs) are often found in proteins that phase separate and can underpin multivalent interactions that help drive phase separation (*Li et al., 2012*). Many phase-separating IDRs are additionally enriched for certain amino acids and are hence referred to as low-complexity domains (LCD) (*Hennig et al., 2015*; *Kato et al., 2012*). Although multiple nuclear events are now being scrutinized through the lens of phase separation, how these processes integrate and communicate with other LLPS or dispersive-state pathways is currently unknown.

Here we report that the *Drosophila melanogaster* replication initiation factors ORC, Cdc6, and Cdt1 possess N-terminal IDRs that facilitate DNA-dependent liquid-liquid phase separation in vitro. Bioinformatic analyses reveal that these initiator IDRs possess a high-complexity sequence signature that is preserved in metazoan homologs, but that is distinct from other condensate-promoting IDRs in other cellular pathways. Biochemical studies show that these IDRs drive the selective co-assembly and enrichment of *Drosophila* initiation factors into liquid phases in the presence of DNA, while simultaneously excluding non-partner proteins that also phase separate. Although *Drosophila* Mcm2-7 does not appear to phase separate on its own, initiation factors can recruit the complex into condensates in a DNA-, ATP-, and ORC/Cdc6/Cdt1-dependent manner. In addition, cellular and genetic studies establish that the Orc1 IDR is critical for its recruitment to mitotic chromosomes and is essential for viability. Collectively, our observations not only reveal a new class of high-complexity sequences that undergo phase separation, but also provide a model for how these elements promote physical interactions between replication initiation factors that could help promote long-range chromosomal communication observed in replication programs.

# Results

## *D. melanogaster* Cdt1 undergoes DNA-dependent liquid-liquid phase separation (LLPS)

The architecture of *D. melanogaster* Cdt1 (*Dm*Cdt1) is markedly different from its budding yeast counterpart (**Figure 1A**). Sequence analysis with the disorder prediction server DISOPRED (*Jones and Cozzetto, 2015*) reveals that the N-terminal sequence of *Dm*Cdt1 is predicted to be an extended intrinsically disordered region (IDR) (**Figure 1A**). Although the sequence of this region is not conserved per se, an N-terminal IDR is present in other metazoan Cdt1s (**Table 1**).

The N-terminal IDR of metazoan Cdt1 contains multiple conserved short linear motifs (SLiMs) (*Davey et al., 2012*) necessary for Cdt1 regulation, such as a PCNA interacting peptide (PIP) box and kinase consensus sequences (*Pozo and Cook, 2016*). However, known SLiMs only account for a small fraction (<25%) of the total length of the metazoan Cdt1 N-terminal IDR, so we asked whether this domain might have more general functional significance. The IDR of *Dm*Cdt1 has a predicted pI that is relatively basic (pI = 10.2); this feature, combined with the presence of tandem WH domains in the protein, suggested that *Dm*Cdt1 might bind DNA. We found that agarose beads coupled with a random 60 bp double-stranded DNA (dsDNA) were able to efficiently pull down *Dm*Cdt1,

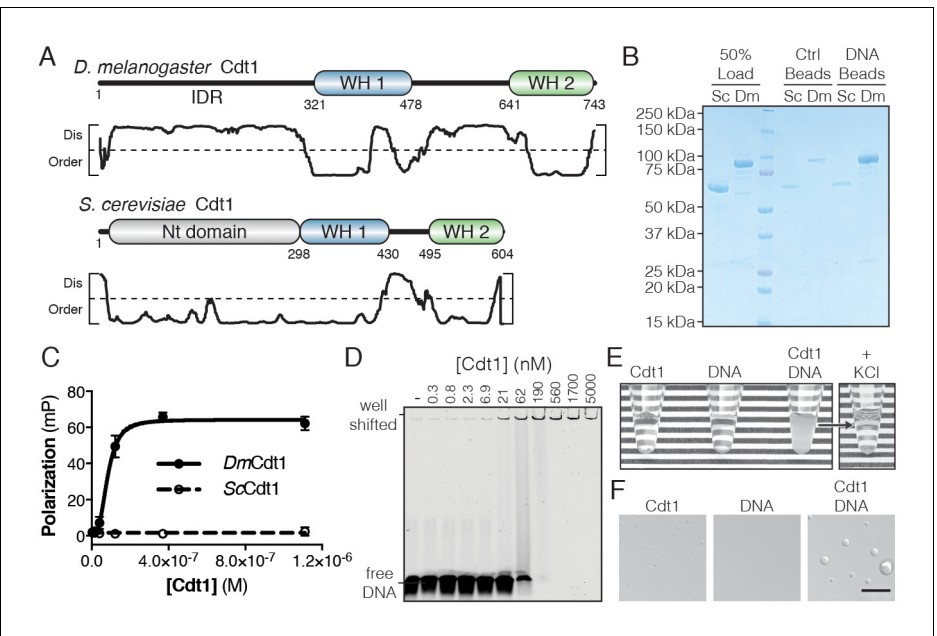

**Figure 1.** *D. melanogaster* Cdt1 undergoes DNA-dependent phase separation. (**A**) Architecture of *D. melanogaster* and *S. cerevisiae* Cdt1. The per-residue DISOPRED (*Jones and Cozzetto, 2015*) disorder prediction score is shown in the plot below each gene, with a cutoff value of 0.5 indicated by the dashed line. Residues scored above this cutoff are predicted to be disordered. (**B**) dsDNA-coupled ('DNA Beads') and control ('Ctrl Beads') agarose beads were used to pull-down *Dm*Cdt1 and *Sc*Cdt1. *Dm*Cdt1 but not *Sc*Cdt1 bound to the DNA-coupled beads. (**C**) Fluorescence anisotropy measurements of Cdt1 binding to a Cy5-labeled duplex oligonucleotide. *Sc*Cdt1 showed no evidence of binding. *Dm*Cdt1 bound with a $K_{d, app}$ = 83 ± 17 nM. (**D**) EMSA analysis of *Dm*Cdt1 binding to duplex DNA. The complex between DNA and *Dm*Cdt1 is heterogeneous and large, and is fully well-shifted at the highest concentrations. In good agreement with anisotropy measurements, the calculated $K_{d, app}$ for *Dm*Cdt1 is ~100 nM. (**E**) Mixing concentrated *Dm*Cdt1 with duplex DNA results in a visible increase in solution turbidity that can be reversed with the addition of KCl. (**F**) DIC microscopy analysis of solutions of Cdt1, DNA, and a Cdt1/DNA mixture (scale bar = 5 µm). Phase-separated droplets were evident when Cdt1 was mixed with DNA. Gel images are representative from three independent experiments.

DOI: https://doi.org/10.7554/eLife.48562.002

The following figure supplement is available for figure 1:

**Figure supplement 1.** Sequence and length-dependence of Cdt1 phase separation.

DOI: https://doi.org/10.7554/eLife.48562.003

**Table 1.** Conservation of initiator IDR sequence features.
Analysis of eukaryotic initiator homologs for the presence of an N-terminal IDR, as well as IDR sequence features ('pI'=isoelectric point; 'FCR'=fraction charged residues).

| | Orc1 | Cdc6 | Cdt1 |
|---|---|---|---|
| *D. melanogaster* | | | |
| N-term IDR length: | 362 aa | 246 aa | 294 aa |
| pI: | 10.2 | 9.4 | 10.1 |
| FCR: | 0.32 | 0.35 | 0.32 |
| Human | | | |
| N-term IDR length: | 300 aa | 136 aa | 175 aa |
| pI: | 10.7 | 10.6 | 10.6 |
| FCR: | 0.33 | 0.29 | 0.29 |
| Mouse | | | |
| N-term IDR length: | 298 aa | 141 aa | 178 aa |
| pI: | 10.2 | 10.0 | 9.8 |
| FCR: | 0.34 | 0.31 | 0.24 |
| *X. laevis* | | | |
| N-term IDR length: | 327 aa | 140 aa | 248 aa |
| pI: | 9.7 | 10.8 | 10.1 |
| FCR: | 0.33 | 0.26 | 0.31 |
| *C. elegans* | | | |
| N-term IDR length: | 242 aa | 172 aa | 194 aa |
| pI: | 9.1 | 9.9 | 10.3 |
| FCR: | 0.39 | 0.38 | 0.35 |
| *D. rerio* | | | |
| N-term IDR length: | 391 aa | 156 aa | 309 aa |
| pI: | 9.8 | 11.1 | 9.9 |
| FCR: | 0.31 | 0.22 | 0.31 |
| *S. cerevisiae* | | | |
| N-term IDR length: | 143 aa | 31 aa | N/A |
| pI: | 4.7 | 6.2 | N/A |
| FCR: | 0.49 | 0.32 | N/A |
| *S. pombe* | | | |
| N-term IDR length: | 117 aa | 133 aa | N/A |
| pI: | 10.6 | 10.2 | N/A |
| FCR: | 0.36 | 0.22 | N/A |

DOI: https://doi.org/10.7554/eLife.48562.004

whereas no such interaction was observed for *Sc*Cdt1 (*Figure 1B*). Quantitative analysis of dsDNA binding using a fluorescence polarization (FP)-based assay also revealed DNA binding by *Dm*Cdt1 ($K_{d, app}$ = 83 ± 17 nM) but not *Sc*Cdt1 (*Figure 1C*). Mouse and *S. pombe* Cdt1 have similarly been shown to bind DNA in vitro (*Houchens et al., 2008*; *Yanagi et al., 2002*).

Electrophoretic mobility assays confirmed DNA binding by *Dm*Cdt1 ($K_{d, app}$ ≈ 100 nM) but also showed that the complex is highly heterogenous (*Figure 1D*). Indeed, elevated concentrations of the protein (>190 nM) resulted in a well-shifted species, suggestive of increasingly higher-order assemblies; however, size-exclusion chromatography showed that purified *Dm*Cdt1 is monodisperse in solution (*Figure 1—figure supplement 1A*). Interestingly, in the course of conducting these studies, we found that upon mixing 20 µM *Dm*Cdt1 with stoichiometric amounts of a 60 bp duplex oligo, the solution became visibly turbid, and that this turbidity was reversible by the addition of salt (400 mM KCl) (*Figure 1E*). Inspection of the turbid solution by differential interference contrast (DIC) microscopy unexpectedly revealed the presence of phase-separated droplets up to 4 µm in diameter (*Figure 1F*). These droplets were fully absent from the DNA alone sample and barely detectable in both number and size in the *Dm*Cdt1 alone sample. Notably, Cdt1 phase separation was dependent on the length, but not the sequence, of the dsDNA substrate, with maximal LLPS occurring in the presence of oligonucleotides longer than 25 basepairs (*Figure 1—figure supplement 1B–C*). To confirm that the affinity determined for Cdt1 in binding DNA was not confounded by the phase transition event, we determined whether a rapid binding/exchange equilibrium was maintained in the FP DNA-binding assay using a competition assay that titrated 'cold' dsDNA against pre-formed Cdt1/

DNA complexes (*Figure 1—figure supplement 1D*). The unlabeled oligonucleotide was able to fully reduce DNA binding by Cdt1 to background levels, with an inhibition constant ($K_{c,\ app}$ = 62 ± 4 nM) comparable to the observed $K_{d,\ app}$ (83 nM), demonstrating that the labeled DNA in droplets is freely exchangeable and therefore in equilibrium between bound and unbound states. Together, these data demonstrate not only that *Dm*Cdt1 binds DNA, but that DNA-binding in turn induces the protein to phase separate.

## DNA-dependent phase separation of *Dm*Cdt1 requires the N-terminal IDR and occurs at physiologic concentrations

The DNA-dependency of Cdt1 phase separation predicts that droplets should be enriched for both protein and nucleic acid. To test this prediction, *Dm*Cdt1 was expressed and purified with an N-terminal enhanced Green Fluorescent Protein (eGFP) tag (eGFP-Cdt1) (*Figure 2A*). A Cy5-labeled duplex oligonucleotide (Cy5-dsDNA) was then mixed with an equimolar amount of protein (5:5 μM) and imaged by two-color fluorescence imaging (*Figure 2B*). When eGFP-Cdt1 was mixed with Cy5-dsDNA, droplets up to 3 μm in diameter appeared, all containing both protein and nucleic acid. Samples with eGFP-Cdt1 alone did not show such droplets, nor were they observed for samples with Cy5-dsDNA only. These data demonstrate that phase separation by Cdt1 is driven, at least in part, by dsDNA-induced coacervation, and further show that dye-labeled oligonucleotides can serve

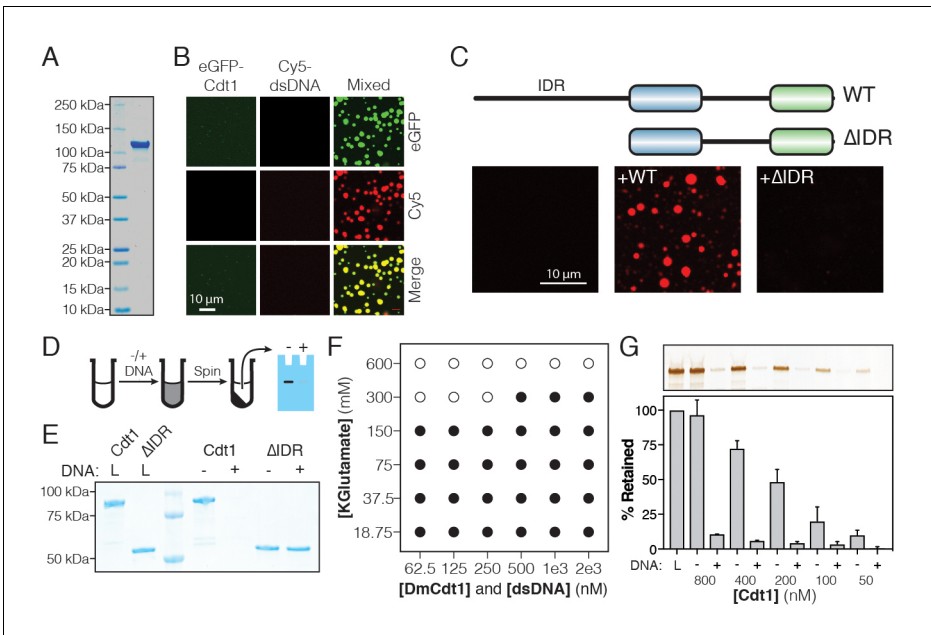

**Figure 2.** *Dm*Cdt1 phase separation is facilitated by an N-terminal IDR. (**A**) SDS-PAGE analysis and Coomassie stain of purified eGFP-Cdt1. (**B**) Samples containing eGFP-Cdt1, Cy5-dsDNA (60 bp), and a mixture of eGFP-Cdt1 and Cy5-dsDNA ('Mixed') were prepared and analyzed by two-color fluorescent microscopy. Droplets were observed in the Mixed sample enriched for both protein (green) and nucleic acid (red). (**C**) Cy5-dsDNA was imaged alone or mixed with either wild-type Cdt1 (WT) or a Cdt1 construct lacking the N-terminal IDR (ΔIDR). Only WT Cdt1 could induce droplet formation. (**D**) Schematic of a condensate depletion assay, a method for assessing phase separation (see Materials and methods for details). (**E**) The depletion assay was utilized to assess the role of the Cdt1 IDR in phase separation. DNA-induced the depletion of WT Cdt1 but not ΔIDR. (**F**) Phase diagram for Cdt1 in the presence of equimolar amounts of sixty basepair dsDNA (filled markers = phase separation observed, unfilled markers = phase separation not observed). (**G**) Depletion assay to assess phase separation at sub-physiological concentrations of Cdt1. DNA-induced phase separation of Cdt1 is seen at the lowest concentration tested (50 nM). Gel images are representative of three independent experiments.

DOI: https://doi.org/10.7554/eLife.48562.005

The following figure supplement is available for figure 2:

**Figure supplement 1.** Properties of *D. melanogaster* Cdt1 phase separation.
DOI: https://doi.org/10.7554/eLife.48562.006

as a proxy for labeled protein when assessing DNA-dependent protein phase separation by fluorescence microscopy.

Given the known importance of protein IDRs in facilitating phase separation (*Mitrea and Kriwacki, 2016*), we hypothesized that the IDR of Cdt1 might likewise be critical for the condensates we observed. We therefore purified an N-terminal IDR deletion of Cdt1 (Cdt1$^{\Delta IDR}$) and determined whether, upon mixing with DNA, a condensed phase could form (*Figure 2C*). No droplets were observed when Cdt1$^{\Delta IDR}$ was mixed with Cy5-dsDNA, even at elevated levels (10 µM protein, 10 µM DNA). To confirm these data with unlabeled DNA, we devised a simple depletion assay (*Figure 2D*). In this approach, protein is incubated with or without DNA, after which the denser, phase-separated material is pelleted by centrifugation, and the degree to which protein is depleted from the supernatant is assessed by SDS-PAGE. In agreement with the microscopy data, no depletion of full-length Cdt1 was observed relative to the load control in the absence of DNA (*Figure 2E*), indicating that no phase separation occurred. Moreover, the full-length Cdt1 signal was fully lost from the supernatant in the presence of DNA, indicating that a near complete partitioning of Cdt1 into the condensed (pelleted) phase took place. When the *Dm*Cdt1 IDR was removed, the protein was again fully retained in the supernatant, regardless of whether DNA was present or not. Notably, a construct containing only the Cdt1 N-terminal IDR residues (Cdt1$^{IDR}$) bound DNA ($K_{d, app}$ = 158 ± 32 nM) and underwent DNA-dependent phase separation on its own (*Figure 2—figure supplement 1A–C*), demonstrating that the Cdt1 IDR is necessary and sufficient for condensation and that it directly interacts with dsDNA. Budding yeast Cdt1, which neither has an IDR (*Table 1*) nor binds DNA (*Figure 1*), did not exhibit phase separation behavior in vitro (*Figure 2—figure supplement 1D–E*). However, all other metazoan Cdt1 homologs possess an N-terminal IDR; tests with human Cdt1 revealed that, similar to *Dm*Cdt1, it also can undergo DNA-induced liquid phase condensation (*Figure 2—figure supplement 1F–G*). Collectively, these data demonstrate the essential role of metazoan Cdt1 IDRs in facilitating DNA-promoted phase separation.

During active replication in the early *D. melanogaster* embryo (0–2 hr), Cdt1 concentration peaks at approximately 70 nM (*Figure 2—figure supplement 1H–J*). Given the low concentration of endogenous Cdt1, it was unclear whether our biochemical experiments, which were conducted at micromolar protein concentrations, would accurately reflect Cdt1 behavior in vivo. To address this question, we used fluorescence microscopy to generate a phase diagram for *Dm*Cdt1, titrating protein and salt concentration, and assaying for the presence or absence of a condensed phase (*Figure 2F*). Droplets were observed down to the lowest concentration of Cdt1 tested (62.5 nM) when salt concentrations were set at or below physiological levels ([KGlutamate]=150 mM). Increasing salt concentration to twice physiological levels (300 mM) shifted the concentration of protein necessary to see phase separation to 500 nM. When the concentration of potassium glutamate was increased to 600 mM, no phase separation was observed for any concentration of Cdt1 tested. The ability of Cdt1 to phase separate at nanomolar concentrations was confirmed by the depletion assay, wherein stoichiometric mixtures of *Dm*Cdt1 and duplex DNA were prepared from 800 nM to 50 nM, and assessed for DNA-dependent depletion of protein in the supernatant (*Figure 2G*). DNA-coupled loss of Cdt1 signal was seen at all concentrations tested. Together, these data demonstrate that *Dm*Cdt1 undergoes DNA-dependent phase separation at physiological protein and salt concentrations.

## ORC and Cdc6 also partition into DNA-dependent liquid phases

Given the role of the *Dm*Cdt1 N-terminal IDR in promoting phase separation, we next examined whether other replication initiation factors might possess similar disordered regions of analogous function. Using DISOPRED (*Jones and Cozzetto, 2015*), we calculated the percent of predicted disordered residues, as well as the longest continuous disordered segment for each *Drosophila* protein required for initiating replication, and ranked them according to low (0–10% predicted unstructured residues), moderate (10–30%), or high disorder (>30%) (*Figure 3A*). In addition to Cdt1, Cdc6 and two subunits of ORC (Orc1 and Orc2) were found to possess a high level of disordered content as a proportion of their total polypeptide chain length, and each contained a continuous region of disorder longer than 200 amino acids. Conversely, the Mcm2-7 subunits, though possessing unstructured regions, contain less than 30% predicted disordered sequence overall, with no single disordered region extending beyond 150 amino acids. Similar patterns were seen for other metazoan replication initiation factors, but not *S. cerevisiae* proteins (*Table 1*).

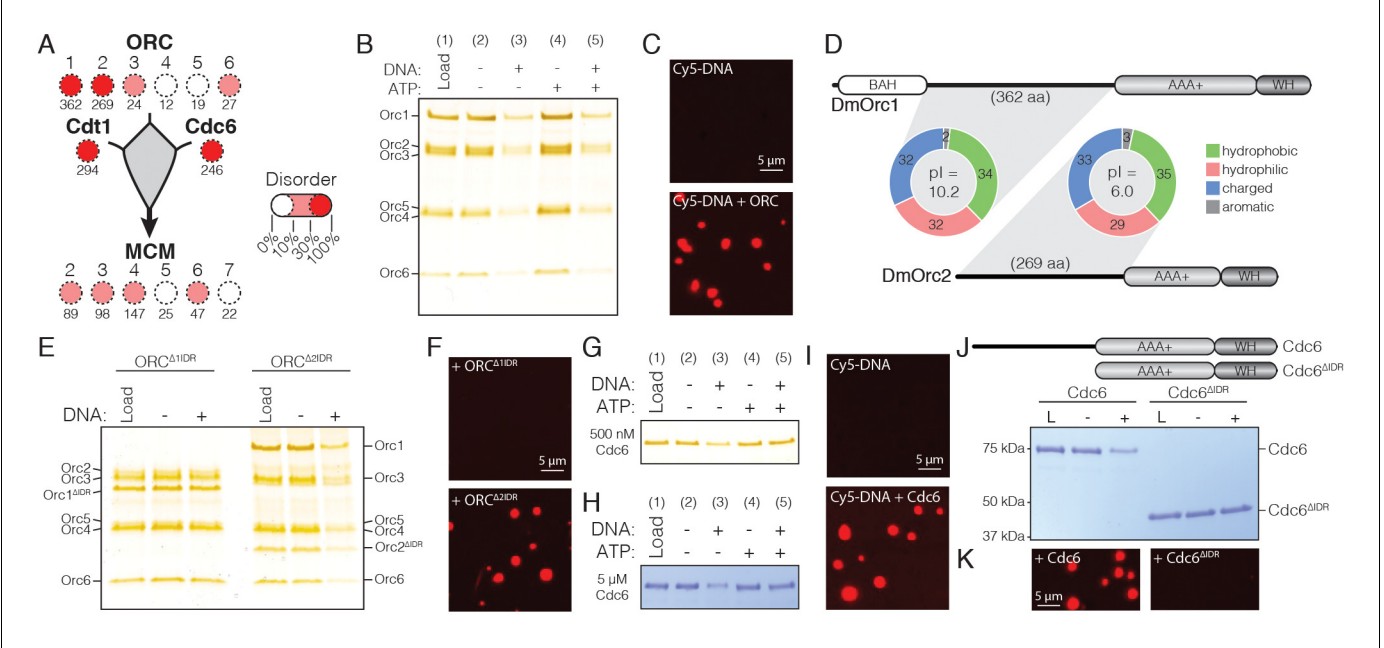

**Figure 3.** *Dm*ORC and *Dm*Cdc6 undergo DNA-dependent phase separation. (**A**) Graphical comparison of the disorder for each *Drosophila* replication initiation factor. Orc1, Orc2, Cdc6, and Cdt1 each contain long IDRs (as denoted by the numbers under each circle) and a high percentage of overall disordered sequence (predicted >30%, depicted by color shading). (**B**) Analysis of ORC phase separation by the depletion assay. ORC (500 nM) phase separates in a DNA-dependent fashion in the presence and absence of ATP. (**C**) Cy5-dsDNA (2.5 μM) was imaged alone and as a mixture with ORC (2.5 μM). In the presence of ORC, large phase-separated droplets formed. (**D**) Two ORC subunits, Orc1 and Orc2, have large N-terminal IDRs. The IDR of Orc1 is longer and is enriched for positively-charged residues. (**E**) Analysis of ORC$^{\Delta 1IDR}$ (500 nM) and ORC$^{\Delta 2IDR}$ (500 nM) phase separation by depletion assay. Loss of the Orc1 IDR abolishes phase separation but loss of the Orc2 IDR has no effect. (**F**) Droplets form when Cy5-dsDNA (2.5 μM) is mixed with ORC$^{\Delta 2IDR}$ (2.5 μM) but not when mixed with ORC$^{\Delta 1IDR}$ (2.5 μM). (**G**) Cdc6 phase separation was assessed by depletion assay at 500 nM and (**H**) 5 μM concentrations. DNA induced Cdc6 phase separation but this was inhibited in the presence of 1 mM ATP. (**I**) Fluorescence imaging reveals phase-separated droplets when Cy5-dsDNA (2.5 μM) is mixed with Cdc6 (20 μM). (**J**) Phase separation analysis for a Cdc6 construct lacking the N-terminal IDR (Cdc6$^{\Delta IDR}$). Cdc6$^{\Delta IDR}$ (500 nM) shows no depletion in the presence of DNA (500 nM). (**K**) Cdc6$^{\Delta IDR}$ (20 μM) is unable to induce droplet formation as assessed by fluorescence microscopy with Cy5-dsDNA (2.5 μM). Gel images are representative of three independent experiments.

DOI: https://doi.org/10.7554/eLife.48562.007

The following figure supplements are available for figure 3:

**Figure supplement 1.** Analysis of ORC and Cdc6 phase separation.

DOI: https://doi.org/10.7554/eLife.48562.008

**Figure supplement 2.** Multiple sequence alignment (MSA) of the Orc1 N-terminal IDR.

DOI: https://doi.org/10.7554/eLife.48562.009

For proteins classified with a high fraction of disordered content (>30%), we next assessed the location of the disordered regions relative to known protein domains (*Figure 3—figure supplement 1A*). In all cases, the longest regions of uninterrupted disorder reside N-terminal to the bulk of the folded domain content of the chain (i.e., upstream of the Cdt1 WH domains and of the AAA+ and WH domains of Orc1, Orc2, and Cdc6). Cdc6 possesses the shortest N-terminal IDR (246 amino acids) and Orc1 the longest (362 residues). There also exist shorter regions of disorder that serve as linker sequences between tandem globular domains, such as between the AAA+ and WH domain of Cdc6 (26 amino acid IDR) or the two WH domains of Cdt1 (152 amino acid IDR).

The realization that Cdc6 and two subunits of ORC possess long N-terminal IDRs suggested that these factors, like Cdt1, might also undergo phase separation in a DNA-dependent fashion, and that this property might promote their functional integration within a single condensed phase. This idea was first tested by examining the ability of recombinant *Dm*ORC to phase separate using the depletion assay (*Figure 3B*, lanes 1–3). Similar to the behavior of *Dm*Cdt1 when it forms condensates, ORC was found to be depleted from the supernatant in the presence, but not in the absence, of DNA. The inclusion of ATP did not have an appreciable effect on phase separation by ORC

(*Figure 3B*, lanes 4–5), and fluorescence microscopy with Cy5-dsDNA confirmed that the DNA-dependent depletion of ORC from the supernatant was due to phase separation (*Figure 3C*, no droplets are observed for DNA alone). Assaying ORC phase separation over a range of protein concentrations (50 nM to 800 nM) consistently revealed ORC depletion when DNA was present but not when it was absent (*Figure 3—figure supplement 1B*). Depletion occurred regardless of oligonucleotide sequence (*Figure 3—figure supplement 1C*), although like Cdt1, maximal phase separation occurred with oligonucleotides longer than 25 basepairs (*Figure 3—figure supplement 1D*). Thus, like Cdt1, ORC is able to undergo LLPS at physiological concentrations in the presence of a variety of DNA substrates without any apparent DNA sequence dependence.

We next investigated whether ORC's ability to undergo phase separation could be targeted to the IDR of a specific subunit, or whether the IDRs of both Orc1 and Orc2 are required for this behavior. In terms of length, the Orc1 IDR, at 362 amino acids, is approximately 100 residues longer than the Orc2 IDR (*Figure 3D*). Comparisons of sequence composition between Orc1 and Orc2 IDRs – performed by calculating the relative percentage of hydrophobic (A, G, I, L, M, P, V), hydrophilic (C, N, S, T, Q), charged (D, E, H, K, R), and aromatic (F, W, Y) residues – show that the IDRs of Orc1 and Orc2 are highly similar in content, with the values for Orc1 and Orc2 within 3% for each amino acid category. Interestingly, Orc1 and Orc2 IDR amino acids are near equally distributed across hydrophobic (36/35%), hydrophilic (30/29%), and charged (32/33%) classes; this preponderance of hydrophobic residues is somewhat unexpected given the predicted unstructured nature of these sequences. Glycine, which is grouped within the hydrophobic class, is often enriched in protein unstructured regions, yet for both Orc1 and Orc2, glycine content is lower than expected for such a region (*Brüne et al., 2018*).

Although the Orc1 and Orc2 IDRs are highly similar in terms of amino acid types, they do show a marked difference in their isoelectric points (pI): the Orc1 IDR (pI = 10.1) is enriched for positively-charged residues (20% positive and 12% negative), whereas the Orc2 IDR (pI = 6.0) is weakly enriched for negatively-charged residues (17% negative and 16% positive). Speculating that this divergence in net charge might be important for facilitating LLPS by ORC, we constructed two mutant *Dm*ORC complexes, one with the Orc1 IDR deleted (ORC$^{1\Delta IDR}$) and the other lacking the Orc2 IDR (ORC$^{2\Delta IDR}$). Phase separation for both constructs was then assessed by the depletion assay and fluorescence imaging with Cy5-dsDNA. ORC$^{1\Delta IDR}$ showed no evidence for phase separation in either assay, whereas the construct lacking the Orc2 IDR exhibited wild-type behavior in both assays (*Figure 3E–F*). These findings demonstrate that *Drosophila* ORC phase separates using interactions that require the N-terminal IDR of Orc1 but not Orc2. Given the predicted lack of structure in the Orc1 IDR we were surprised to discover that this region is relatively well-conserved across the *Drosophila* genus (67% identity, *Figure 3—figure supplement 2*) and that the composition, length, and pI of this region is conserved across the metazoan phyla (*Table 1*). Although *S. cerevisiae* Orc1 possesses an N-terminal IDR, this region is shorter than observed in metazoa and also has an acidic pI (*Table 1*). Consistent with this observation, budding yeast ORC does not phase separate (*Figure 3—figure supplement 1E*).

We next asked whether Cdc6 can undergo DNA-dependent LLPS. Similar to budding yeast Cdc6 (*Feng et al., 2000*), *Dm*Cdc6 was found to interact with DNA by dsDNA-coupled agarose bead pull-down assays; the presence of ATP had no significant effect on Cdc6 DNA-binding (*Figure 3—figure supplement 1F*). Upon testing whether Cdc6 phase separates in the presence of DNA using the depletion assay (*Figure 3G*), we observed a reproducible DNA-dependent depletion of Cdc6 from the supernatant (*Figure 3G* lanes 1–3); however, in contrast to ORC, this depletion was inhibited by the presence of ATP (*Figure 3G* lanes 4–5). A recent report that ATP can function as a hydrotrope to enhance intracellular protein solubility may provide an explanation for its effect on Cdc6 phase separation (*Patel et al., 2017*). Repeating the assay at higher concentrations of *Dm*Cdc6 shows that unlike *Dm*Cdt1, which appeared to fully partition into phases at higher concentrations (*Figure 2E*), Cdc6 was only partially depleted from the supernatant (*Figure 3H*). When examined over a range of protein concentrations and DNA sequences (*Figure 3—figure supplement 1G–H*), Cdc6 consistently showed phase separating behavior, although the protein never completely partitioned into the pellet. Notably, Cdc6 underwent complete partitioning into the condensed phase in the presence of plasmid DNA, whereas only partial depletion was observed with short oligonucleotides (*Figure 3—figure supplement 1I*), indicating that for *Dm*Cdc6, LLPS requires longer DNA segments than either *Dm*Cdt1 or *Dm*ORC. Fluorescence microscopy with Cy5-dsDNA confirmed that Cdc6

forms condensates in these conditions (*Figure 3I*), while tests with a Cdc6 mutant lacking the N-terminal IDR (Cdc6$^{\Delta IDR}$) (*Figure 3J*) showed that this construct has no capacity to phase separate (*Figure 3J–K*). Collectively, these studies confirm that *Drosophila* Cdc6 can form DNA-dependent condensates in a manner that requires its N-terminal IDR.

## Initiator IDRs co-assemble and recruit Mcm2-7

The independent ability of ORC, Cdc6, and Cdt1 to phase separate, as well as their reliance on the same structural feature for this activity (an N-terminal IDR), suggested that these regions might allow initiators to co-partition into a single condensed phase. To test this idea, condensates were prepared by mixing ORC with Cy5-dsDNA, after which RFP-tagged *Dm*Cdc6 (tRFP-Cdc6) and eGFP-Cdt1 were sequentially added. Both tRFP-Cdc6 and eGFP-Cdt1 showed marked enrichment within ORC droplets (*Figure 4A*). Moreover, although phase separation by Cdc6 was partially inhibited by ATP, all three initiator components can co-associate in a liquid phase in the presence of ATP (*Figure 4A*). Thus, while ATP appears to impede phase separation by Cdc6 on its own, the presence of the other initiators overcomes this barrier and allows the full suite of helicase loading factors to co-localize into a highly concentrated protein/nucleic acid-rich phase.

Initiator IDRs have a distinct amino acid sequence signature compared to other phase-separating factors such as human FUS and EWS, or *C. elegans* Laf1 (*Figure 4—figure supplement 1A–E*). Generally speaking, initiator IDRs can be characterized as having: 1) a high degree of sequence complexity, and approximately equal representation of charged, hydrophilic, and hydrophobic residues, 2) a deficiency in aromatic amino acids and glycine, and 3) a net positive charge. This sequence pattern suggested that initiator IDRs might serve as specificity determinants for forming condensates. This hypothesis was tested by comparing the recruitment of eGFP-tagged human FUS (eGFP-FUS) and eGFP-*Dm*Cdt1 into preformed ORC/Cy5-dsDNA droplets. Relative to the bulk phase, a 10-fold enrichment of eGFP-Cdt1 signal intensity is observed within ORC/Cy5-dsdNA droplets (*Figure 4B–C*). Conversely, no enrichment is observed for eGFP-FUS (*Figure 4B–C*). Thus, initiator condensates show a degree of specificity that allows for exclusion of other proteins with non-congruent IDRs.

Given the liquid phase-forming properties of ORC, Cdc6, and Cdt1 — along with their enzymatically coordinated role in loading the Mcm2-7 replicative helicase onto DNA — we asked whether *Drosophila* Mcm2-7 (*Dm*Mcm2-7) might undergo LLPS as well. We first addressed this question using a modified form of the depletion assay where we assessed partitioning of FLAG-tagged Mcm2-7 to both the dilute and condensed phases by Western blotting (*Figure 4D*). No evidence for phase separation by Mcm2-7 alone was observed under these conditions either in the presence or absence of ATP (*Figure 4E–F*). Although specific MCM subunits contain predicted disordered domains (*Figure 3A*), these regions are in all cases shorter than the N-terminal IDRs of ORC, Cdc6 and Cdt1, and have a different amino acid composition (*Figure 4—figure supplement 2A–B*). The IDRs of Mcm2 and Mcm3, for example, have an acidic pI (pI <6), and although Mcm4 contains an N-terminal IDR with a basic pI (pI = 11.0), this region has fewer charged residues compared to the other initiator IDRs. Fluorescence microscopy with FLAG-Mcm2-7 and Cy5-dsDNA showed no evidence of droplet formation (*Figure 4—figure supplement 2C*), confirming that Mcm2-7 does not undergo phase separation.

Mcm2-7 recruitment to and stable loading on DNA requires the formation of a pre-Replicative Complex (pre-RC) containing ORC, Cdc6, and Cdt1 (*Bowers et al., 2004*; *Evrin et al., 2013*; *Kang et al., 2014*). We therefore asked whether *Dm*Mcm2-7 might be recruited into condensed phases in either the presence of the initiators alone, or also with ATP, which is required for loading. Depletion reactions were set up that contained Mcm2-7 and DNA, and supplemented with either ATP or *Drosophila* ORC, Cdc6, and Cdt1 ('OCC'), or with both ATP and the OCC (*Figure 4G*). All Mcm2-7 recruitment reactions included plasmid DNA, as plasmid most effectively induces phase separation *Figure 1—figure supplement 1B–C* and *Figure 3—figure supplement 1C–D,H–I*), is sufficiently long to accomadate Pre-RC assembly, and is capable of trapping loaded Mcm2-7 (which is known to slide off linear DNA ends; *Evrin et al., 2009*). No appreciable pelleting of Mcm2-7 alone was observed in either the presence or absence of ATP, or with just the OCC; however, when ATP was included with the OCC, an ~8 fold enrichment of Mcm2-7 was observed in the pelleted phase. These data demonstrate that although *Dm*Mcm2-7 does not phase separate on its own, it can be specifically recruited into condensates when initiation factors and ATP are present.

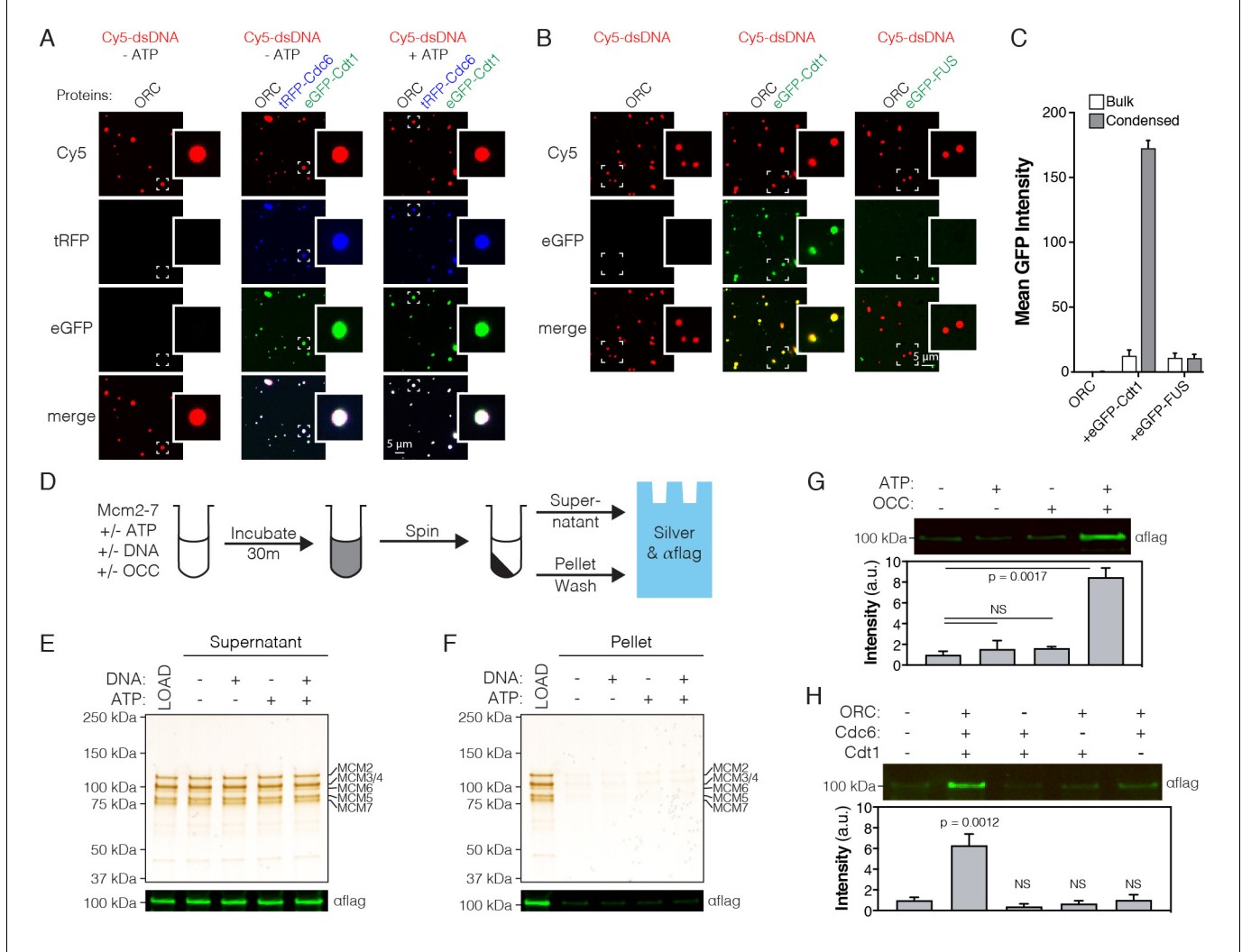

**Figure 4.** ATP-dependent recruitment of *Dm*Mcm2-7 into liquid phases containing DNA, *Drosophila* ORC, Cdc6 and Cdt1. (**A**) Analysis of tRFP-Cdc6 (2.5 µM) and eGFP-Cdt1 (2.5 µM) recruitment to pre-formed ORC/Cy5-dsDNA (2.5/2.5 µM) droplets in the absence ('-ATP') and presence ('+ATP') of ATP (1 mM). (**B**) eGFP-Cdt1 (500 nM) or eGFP-FUS (500 nM) was added to reactions and assessed for the ability to co-localize with preformed ORC/ Cy5-dsDNA droplets (2.5/2.5 µM). In samples containing eGFP-Cdt1, all ORC/Cy5-dsDNA droplets are enriched for eGFP signal. No enrichment is observed in samples containing eGFP-FUS.). (**C**) Quantitation of eGFP signal intensity within and outside of ORC/Cy5-dsDNA droplets for samples in panel (B). (**D**) Schematic of a depletion assay to assess Mcm2-7 phase partitioning; both the supernatant and pellet were assessed for either the loss or enrichment, respectively, of Mcm2-7. (**E**) Mcm2-7 (500 nM) was not depleted from the dilute phase and, consistently, was absent from the condensed phase (**F**). (**G**) Loading reactions were prepared that contained either ORC, Cdc6, and Cdt1 (OCC) or ATP, or both the OCC and ATP, and phase separation of Mcm2-7 was assessed by the depletion assay. In the presence of both the OCC and ATP, Mcm2-7 is significantly enriched in the condensed phase. (**H**) Loading reactions were set up that contained both ATP and DNA, but from which individual initiators were removed (ORC, Cdc6, or Cdt1), and the depletion assay performed. In the absence of any one initiator, Mcm2-7 no longer partitions to the pelleted phase.

DOI: https://doi.org/10.7554/eLife.48562.010

The following figure supplements are available for figure 4:

**Figure supplement 1.** Replication initiation factors form a new class of phase separating IDRs.
DOI: https://doi.org/10.7554/eLife.48562.011

**Figure supplement 2.** Analysis of *D. melanogaster* Mcm2-7 IDRs.
DOI: https://doi.org/10.7554/eLife.48562.012

Because it was possible that some non-specific, but nevertheless ATP-dependent, feature of the condensates might adventitiously pull Mcm2-7 complexes into the pelleted phase, we next investigated whether ORC, Cdc6, and Cdt1 are all required for partitioning. *Dm*Mcm2-7 depletion reactions were repeated with both ATP and DNA, except that this time individual initiators were

withheld (*Figure 4H*). Whereas partitioning of Mcm2-7 into the condensed phase is seen with the intact OCC, no enrichment of the helicase complex is observed when any one of the initiation proteins is omitted from the reaction (*Figure 4H*). Collectively, these data demonstrate that Mcm2-7 does not phase separate on its own, but that it can be selectively enriched in condensed phases containing all of the factors required for helicase loading (ORC, Cdc6, and Cdt1) and ATP.

## The association of ORC with chromatin in vivo is not dependent on its ATPase function but does require the Orc1 N-terminal IDR.

The data presented thus far show that *D. melanogaster* ORC, utilizing the Orc1 N-terminal IDR, can partition with DNA into phase condensates regardless of whether ATP is present or not. However, previous work has demonstrated that the tight binding of DNA by yeast and metazoan ORC is dependent upon ATP binding to its Orc1 subunit (*Bell and Stillman, 1992*; *Bleichert et al., 2018*; *Chesnokov et al., 2001*; *Giordano-Coltart et al., 2005*). This dependency likely reflects the ability of ORC to encircle DNA within its central channel (*Bleichert et al., 2018*; *Li et al., 2018*; *Yuan et al., 2017*). A discrete patch within the *Dm*Orc1 IDR that is close to the protein's ATPase domain has been demonstrated as necessary but not sufficient for the ATP-dependent binding of DNA by ORC in vitro (*Bleichert et al., 2018*). However, whether ATP-independent DNA interactions, such as those participating in LLPS, might be important for the recruitment of ORC to chromatin in cells has not been established.

To first assess the need for ATP-binding by ORC to promote chromatin association in vivo, we examined the dynamics of this interaction in live *D. melanogaster* embryos. A transgenic fly line expressing eGFP-tagged Orc1 under the control of its endogenous promoter was made, and the functional integrity of this transgene confirmed by its ability to rescue a null allele (*Table 2*). In parallel, we constructed a mutant Orc1 transgene, eGFP-Orc1$^{WalkerAB}$, that contains mutations in both its Walker A (K604A) and Walker B (D684A/E685A) ATPase motifs; prior biochemical studies have shown that these changes interfere with ATP binding and hydrolysis, respectively, in both budding yeast and flies (*Chesnokov et al., 2001*; *Klemm et al., 1997*; *Klemm and Bell, 2001*). Genetic crosses revealed that eGFP-Orc1$^{WalkerAB}$ was unable to rescue the null allele (*Table 2*). In principle, the incapacitated function of the eGFP-Orc1$^{WalkerAB}$ transgene could arise either from an inability to localize to chromatin, or from downstream replication initiation events that require ATP turnover. To test whether the Orc1$^{WalkerAB}$ construct showed altered chromatin recruitment, we directly visualized the dynamics of eGFP-Orc1 chromatin association in embryos with RFP-tagged histone H2A by lattice light-sheet microscopy (*Chen et al., 2014*). Previous work with a transgenic fly line expressing an eGFP-tagged Orc2 subunit has demonstrated that in the early embryo, ORC associates with anaphase chromosomes (*Baldinger and Gossen, 2009*). Consistent with this observation, we observed clear loading of eGFP-Orc1 onto chromosomes beginning in anaphase, with a 2.7 (±0.4)-fold chromosomal enrichment of eGFP-Orc1 (*Figure 5A,C–D* and *Figure 5—video 1*). Interestingly, eGFP-Orc1$^{WalkerAB}$ also showed wild-type like chromatin association (*Figure 5B–D* and *Figure 5—video 2*, 2.7 (±0.3)-fold enrichment). These data demonstrate that the initial association of ORC with chromatin in vivo proceeds through a DNA-binding mode that does not require ATP.

Other aspects of metazoan ORC are also thought to play a role with how it associates with chromosomes. These include recruitment to histone H4K20me2 marks through an N-terminal Bromo-Adjacent Homology (BAH) domain found in Orc1 (*Kuo et al., 2012*; *Noguchi et al., 2006*), and the direct binding of DNA by the Orc6 TFIIB domains (*Balasov et al., 2007*; *Liu et al., 2011*). To more

**Table 2.** Rescue of Orc1 null allele (orc1$^{4739}$) by WT Orc1 and WalkerAB mutant transgenes.

|  | orc1$^{4739}$/Cyo; tg/tg | orc1$^{4739}$/Cyo; tg/TM3 | orc1$^{4739}$/orc1$^{4739}$; tg/tg | orc1$^{4739}$/orc1$^{4739}$; tg/TM3 |
|---|---|---|---|---|
| WT eGFP-Orc1 | 223♀<br>209♂ | 347♀<br>428♂ | 86♀<br>80♂ | 196♀<br>186♂ |
| eGFP-Orc1$^{WalkerAB}$ | 413♀<br>396♂ | 622♀<br>847♂ | 0♀<br>0♂ | 0♀<br>0♂ |
| eGFP-Orc1$^{ΔIDR}$ | 360♀<br>365♂ | 617♀<br>655♂ | 0♀<br>0♂ | 0♀<br>0♂ |

DOI: https://doi.org/10.7554/eLife.48562.015

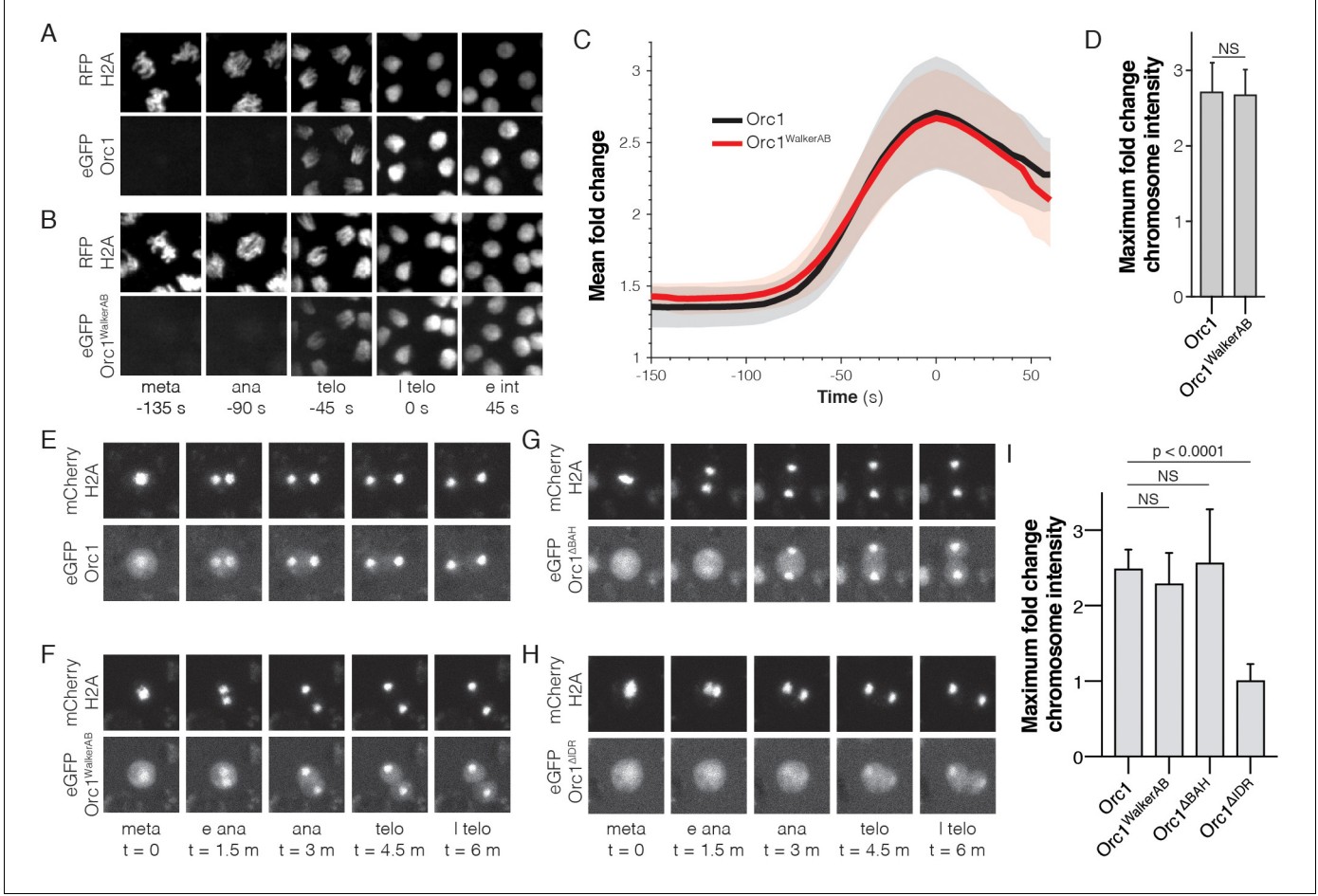

**Figure 5.** The Orc1 N-terminal IDR, but not ATP binding, is required for chromatin recruitment. (**A**) Representative maximal intensity projections of time series images of a mitotic event in *D. melanogaster* embryos expressing eGFP-Orc1 and His2A-RFP ('meta'=metaphase, 'ana'=anaphase, 'telo'=telophase, 'l telo'=late telophase, and 'e int'=early interphase). Loading of eGFP-Orc1 onto chromosomes reached a maximum in late telophase (t = 0). (**B**) As in (**A**), but with eGFP-Orc1$^{WalkerAB}$ and His2A-RFP embryos. (**C**) Quantitative analysis of the fold-change in eGFP signal intensity on chromosomes as cells progress through mitosis. Time is registered with respect to late telophase where maximum loading was observed. (**D**) No difference in maximum fold intensity was observed between eGFP-Orc1 and eGFP-Orc1$^{WalkerAB}$. (**E–H**) Analysis of Orc1 chromatin association in S2 cells transiently transfected with mCherry-His2A and either wild-type Orc1 (**E**), Orc1$^{WalkerAB}$ (**F**), Orc1$^{\Delta BAH}$ (**G**), or Orc1$^{\Delta IDR}$ (**H**) ('meta'=metaphase, 'e ana'=early anaphase, 'ana'=anaphase, 'telo'=telophase, and 'l telo'=late telophase). (**I**) Quantitative analysis of the fold-change in eGFP signal intensity observed on telophase chromosomes for each of the Orc1 constructs (**E–H**). No significant difference was observed between WT, Orc1$^{WalkerAB}$, and Orc1$^{\Delta BAH}$, but Orc1$^{\Delta IDR}$ was not recruited to chromosomes.

DOI: https://doi.org/10.7554/eLife.48562.013

The following video and figure supplement are available for figure 5:

**Figure supplement 1.** Dynamics of Orc1 chromosome recruitment in tissue culture cells and embryos.

DOI: https://doi.org/10.7554/eLife.48562.014

**Figure 5—video 1.** Mitotic dynamics of eGFP-Orc1. eGFP-Orc1 (green) loads onto chromosomes (His2A-RFP, white) beginning in anaphase and reaches a maximum in telophase.

DOI: https://doi.org/10.7554/eLife.48562.016

**Figure 5—video 2.** Mitotic dynamics of eGFP-Orc1$^{WalkerAB}$.

DOI: https://doi.org/10.7554/eLife.48562.017

**Figure 5—video 3.** Example of chromatin segmentation from the His2A-RFP signal.

DOI: https://doi.org/10.7554/eLife.48562.020

rapidly sort through different Orc1 mutants and their role in chromatin association, we turned to *D. melanogaster* S2 tissue culture cells. A series of eGFP-labeled Orc1 constructs to be used for co-expression analysis were constructed, along with a mCherry-tagged variant of histone H2A to provide a benchmark for timing events of interest. To confirm that the S2 cell culture system recapitulates the behavior of ORC seen in embryos, the dynamics of wild-type Orc1 in mitotic cells were imaged first. Consistent with our observations in embryos, Orc1 loaded onto chromosomes beginning in anaphase (2.5 ± 0.2 fold enrichment; *Figure 5E,I*), and mutations within the Orc1 Walker A and B motifs had no effect on the observed recruitment (2.3 ± 0.4 fold enrichment, *Figure 5F,I*). A time course analysis of Orc1 enrichment reveals that loading onto chromosomes begins in anaphase and reaches peak intensity in telophase (*Figure 5—figure supplement 1A*). We next tested how deletion of either the Orc1 BAH domain (Orc1$^{\Delta BAH}$) or the Orc1 IDR (Orc1$^{\Delta IDR}$) altered Orc1 chromatin recruitment in S2 cells (the design of these constructs retained the nuclear localization signal present in the most N-terminal portion of the Orc1 IDR, which directly follows the BAH domain). The Orc1$^{\Delta BAH}$ construct showed wild-type like association with chromosomes (2.6 ± 0.7 fold enrichment; *Figure 5G,I*), demonstrating that the BAH domain is not required for chromatin association in vivo. By contrast, chromosome recruitment of Orc1$^{\Delta IDR}$ was fully abolished to background levels (1.0 ± 0.2 fold enrichment; *Figure 5H–I*), such that no increase in ORC signal on chromosomes was observed as cells progressed through mitosis. A closer inspection of the data reveals that the chromosome signal for Orc1$^{\Delta IDR}$ appears to decrease upon entry into anaphase but recovers to background (metaphase) levels in telophase (*Figure 5—figure supplement 1A*, red line). Finally, we used fly genetics to confirm the essential function of the Orc1 N-terminal IDR, demonstrating that an Orc1$^{\Delta IDR}$ transgene is incapable of rescuing an Orc1 null allele (*Table 2*). Currently, we know of no other function of ORC that is essential for viability other than its activities in DNA replication. Together, these data demonstrate that the Orc1 N-terminal IDR is both essential for viability and facilities the ATP-independent recruitment of ORC to mitotic chromosomes.

## Initiator phase separation is regulated by CDK-dependent phosphorylation

Our analysis of Orc1 dynamics in cells shows that the association of the complex with chromatin is regulated and does not occur until anaphase (*Figure 5*). This result suggests that a cell-cycle-dependent change in the status of either chromatin or ORC might be responsible for promoting DNA binding. Previous work has shown that the association of human and *X. laevis* ORC with chromatin is regulated by CDK-dependent phosphorylation (*Findeisen et al., 1999*; *Lee et al., 2012*; *Li et al., 2004*; *Rowles et al., 1999*). Interestingly, the major sites for CDK action map to the Orc1 IDR and the phosphorylation of these loci interferes with the binding of *Drosophila* ORC to DNA in vitro (*Remus et al., 2005*); the binding of *D. melanogaster* ORC to mitotic chromosomes also requires cessation of CDK activity (*Baldinger and Gossen, 2009*). Despite these observations, the mechanism by which phosphorylation of ORC inhibits its ability to bind chromatin has remained unclear.

Since phase separation by replication initiation factors appears coupled to electrostatic interactions between their IDRs and DNA (*Figure 2—figure supplement 1*), as indicated by the salt-sensitivity of phase separation (*Figure 2F*) and the basic nature of the initiator IDRs (*Table 1*), we hypothesized that initiator phosphorylation might inhibit the ability of these factors to form condensates. An analysis of putative CDK phosphorylation sites shows that Orc1 hosts sixteen such motifs (*Figure 6A*), all but one of which localizes to the Orc1 IDR. Of these, seven IDR sites represent full CDK consensus sequences, [S/T]PX[K/R], while the others represent the minimal sequence, [S/T]P. Given the abundance of phosphorylation sites within the Orc1 IDR, we asked whether the Cdc6 and Cdt1 IDRs are also enriched with CDK consensus sequences (*Figure 6A*). Cdc6 and Cdt1 have seven and sixteen putative sites, respectively. For Cdc6, all sites reside within the N-terminal IDR and four represent full consensus sequences. For Cdt1, 13/16 sites are within the N-terminal IDR and nine represent the full consensus sequence. Thus, replication initiation factor IDRs, which constitute the regions responsible for phase separation, retain the vast majority of CDK phosphorylation sites.

To test the functional implications of initiator phosphorylation, we modified our depletion assay to include pre-treatment with CDK/Cyclin. Initiation factors were treated for 60 min with recombinant CDK1/CycA expressed and purified from insect cells. After treatment, DNA was added to induce phase separation and the depletion assay was completed as above, with phase separation assessed by protein removal from the supernatant after centrifugation (*Figure 6B*). Reactions were

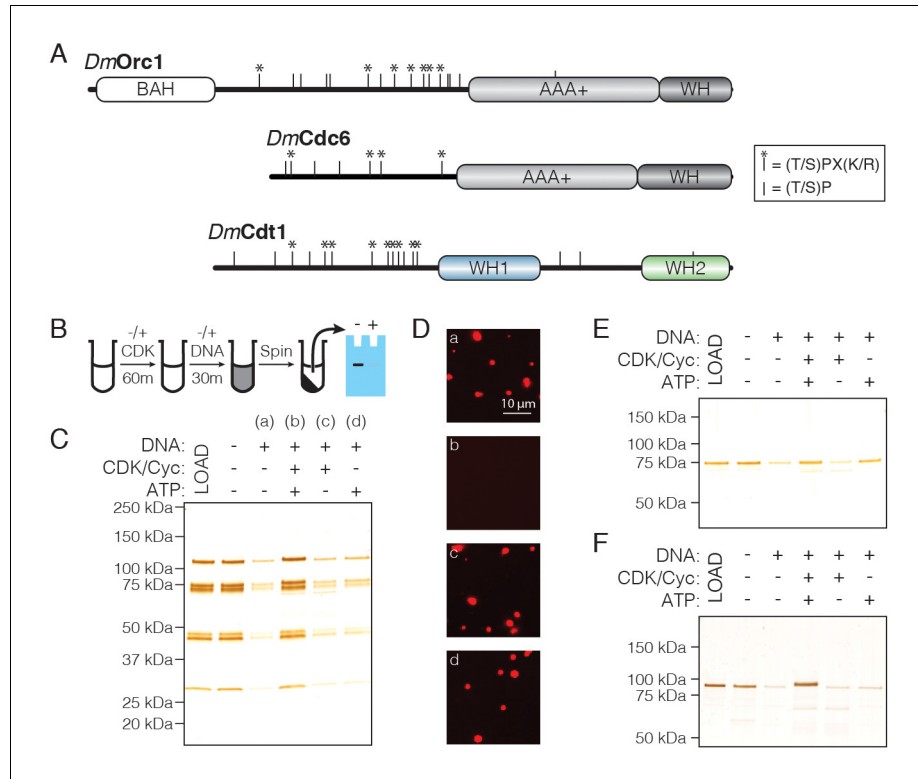

**Figure 6.** CDK/Cyclin-dependent phosphorylation of initiators regulates phase separation. (A) Schematic of CDK/
Cyc phosphorylation consensus sequences across the *Dm*Orc1, *Dm*Cdc6, and *Dm*Cdt1 proteins. Tic marks along
each sequence represent the minimum consensus sequence ([T/S]P); those denoted with an asterisk (*) indicate
the full consensus sequence ([T/S]PX[K/R]). (B) Schematic of the depletion assay used to assess the effect of
phosphorylation on phase separation. (C) Depletion assay with untreated ORC or ORC treated with CDK1/CycA/
ATP, CDK1/CycA alone, or ATP alone. No depletion is observed when ORC is pre-treated with CDK1/CycA/ATP,
but either reagent alone has no effect. (D) Fluorescence microscopy analysis of samples in (C) confirms that
phosphorylated ORC is unable to induce phase separation ('a', untreated ORC; 'b', ORC treated with CDK1/CycA/
ATP; 'c', ORC treated with CDK1/CycA; 'd', ORC treated with ATP). (E) Analysis of Cdc6 depletion under
conditions described in (C). Cdc6 phase separation is inhibited in the presence of CDK1/CycA/ATP, as well as by
ATP alone. (F) Analysis of Cdt1 depletion under conditions described in (C). Cdt1 phase separation is fully
inhibited when treated with CDK1/CycA/ATP. A slight decrease in Cdt1 mobility is seen in the phosphorylated
reaction.
DOI: https://doi.org/10.7554/eLife.48562.018

set up that lacked either CDK1/CycA or ATP to control for the independent effect of each reagent.
DNA was again seen to robustly stimulate phase separation by ORC, and the addition of either
CDK1/CycA or ATP alone had no effect on this behavior; however, treatment with CDK1/CycA and
ATP fully inhibited ORC phase separation (*Figure 6C*). Analysis of the reactions by light microscopy
corroborated the results of the depletion assay (*Figure 6D*), demonstrating that ORC phase separation
is inhibited by CDK-dependent phosphorylation. We next asked whether Cdc6 and Cdt1 are
similarly regulated. Although Cdc6 phase separation is inhibited in the presence of both CDK1/CycA
and ATP (*Figure 6E*), it is likewise inhibited by ATP alone and thus a specific effect of phosphorylation
could not be assessed. By comparison, phase separation by Cdt1 was fully inhibited by treatment
with CDK1/CycA and ATP, but not by either CDK1/CycA or ATP alone (*Figure 6F*). These
results demonstrate that CDK/Cyclin phosphorylation can directly inhibit liquid phase condensation
by replication initiation factors.

## Discussion

We report here the discovery of a novel condensation property for metazoan replication initiator proteins that directly impacts their biochemical and cellular functions. These findings have substantial implications for the mechanisms of DNA replication initiation, and for understanding how replication is managed across a topologically complex chromatin substrate. More generally, our findings also expand the catalogue and types of IDRs that are known to drive protein phase separation and reinforce the notion that such elements can act as a 'sorting code' for distinguishing partner proteins from other factors in the cell.

### Recruitment of ORC to chromosomes

Evolution has created multiple mechanisms for recruiting replication initiation factors to chromosomes (reviewed in *Bleichert et al., 2017*). The prevailing view is that recruitment occurs principally through the ability of the origin recognition complex (ORC) to bind origin DNA in an ATP-dependent manner that depends on DNA encirclement within the central channel of the complex (*Bleichert et al., 2018*; *Li et al., 2018*; *Sun et al., 2013*; *Yuan et al., 2017*). However, we demonstrate here that in the absence of ATP, *D. melanogaster* ORC can interact with DNA at physiological protein and salt concentrations in vitro (*Figure 3*), and that the Orc1 ATP-binding and hydrolysis motifs are dispensable for its recruitment to chromatin in vivo, both in the early embryo and in cell culture. (*Figure 5*). The N-terminal intrinsically disordered region (IDR) of *Dm*Orc1 is shown to be the key element underpinning the ATP-independent association of ORC with chromosomes (*Figure 3* and *Figure 5*). Corroborating these results is previous work that demonstrates a specific role for human and *C. elegans* Orc1 in facilitating ATP-independent ORC/DNA binding (*Giordano-Coltart et al., 2005*; *Kara et al., 2015*; *Sonneville et al., 2012*; *Vashee et al., 2003*) and a direct role for a short DNA-binding motif within the Orc1 IDR (*Bleichert et al., 2018*; *Kawakami et al., 2015*; *Li et al., 2018*). It remains to be determined whether the Orc1 IDR alone is sufficient for proper origin recognition (as opposed to general chromatin recruitment), or whether the action of this element works in concert with other DNA-interaction surfaces of ORC, such as the Orc1 BAH domain (*Kuo et al., 2012*) or the TFIIB domains of Orc6 (*Liu et al., 2011*).

The long-standing observation that DNA binding by ORC is enhanced by ATP (*Bell and Stillman, 1992*; *Chesnokov et al., 2001*; *Vashee et al., 2003*) would at first seem at odds with the ATP-independent DNA association phenomena reported here. This dichotomy can be reconciled by invoking a two-step model wherein (for metazoan ORC at least) the tight, ATP-dependent encirclement of a short segment of DNA occurs after an initial set of weaker interactions takes place between an initiator IDR and a DNA segment or chromatin region (). These dynamic interactions would likely involve contacts between short basic motifs within the Orc1 IDR and DNA, and between the IDRs of ORC molecules as well. The association of ORC with relatively small, diffusible, and nucleosome-free DNAs in this fashion can directly lead to phase separation in vitro, as observed here. By contrast, multivalent ORC-ORC and ORC-DNA interactions in vivo need to contend with a much larger and less pliable chromatin substrate. The nature of the DNA substrate in this context would be expected to resist large-scale condensation by ORC-ORC and ORC-DNA interactions; these associations would instead be expected to help distribute ORC across chromosomes in a manner akin to surface 'wetting,' a well-known property of condensate-forming systems (*Brangwynne et al., 2009*; *Feric et al., 2016*).

The initial IDR-dependent recruitment of ORC to DNA would poise the complex for the second binding step whereby ORC encircles a nucleic acid duplex using specialized and structurally-defined DNA binding elements that line the ORC central channel. These interactions would rely on a conformational state that is stabilized by ATP binding, rendering the complex competent for catalyzing Mcm2-7 loading. Consistent with this two-step model, an eGFP-tagged Orc1 transgene carrying an ATP binding and hydrolysis mutation loses its ability to complement a null allele of the protein (*Table 2*), even though it associates with chromatin in what appears to be a wild-type manner (*Figure 5*). Although the precise link between the association of ORC with chromatin and the specification of a particular locus as a *bona fide* origin is unclear, it is tempting to speculate that some ATP-controlled aspect of DNA encirclement underpins this step.

The potential for a link between IDR-dependent modes of DNA-binding and higher-order DNA structure/origin organization remains to be established. It will be particularly important to

understand whether chromatin or DNA structural elements implicated in origin selection (e.g., G-quadruplex sequences; *Cayrou et al., 2015*; *Cayrou et al., 2012*) might play a role in initiator phase separation. Our finding that condensation in vitro is neither sequence-specific nor possible with short DNA fragments ($\leq$ 25 basepairs, *Figure 3—figure supplement 1*) is consistent with the known absence of sequence-defined origins in metazoan species (*Heinzel et al., 1991*; *Hyrien and Méchali, 1993*; *Méchali and Kearsey, 1984*) and a lack of sequence specificity observed for metazoan ORCs in binding to DNA in vitro (*Remus et al., 2004*; *Vashee et al., 2003*). Whether the DNA length requirement for LLPS by ORC might impact the selection of origin sites is currently unknown, but may influence the preference of ORC for open, nucleosome-free regions of chromatin (*MacAlpine et al., 2010*; *Miotto et al., 2016*).

## Formation of the Pre-RC and helicase loading

An important property of initiator IDRs is that they allow for both self-self and partner protein interactions (*Figure 4A*). Thus, as ORC forms local interactions with itself to spread out along chromatin, this condensation would in turn be expected to promote the formation of stable subunit-subunit interactions that are ultimately necessary for helicase loading (such as the proper docking of Cdc6 into the ORC ring, ). Importantly, this mechanism of co-association is seen to predominate in vitro at

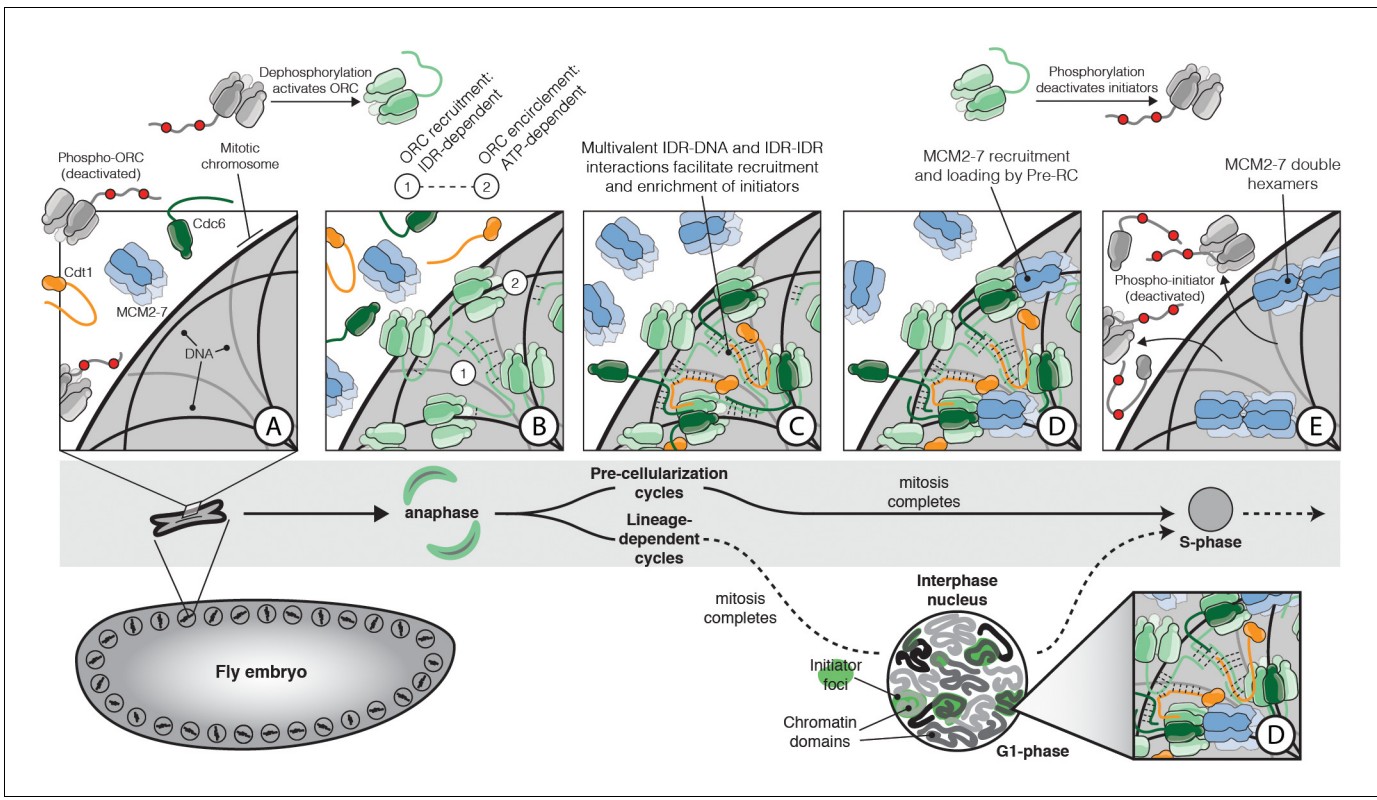

**Figure 7.** The role of IDR- and DNA-dependent initiator co-association in replication initiation. Prior to mitosis, ORC exists in an inhibited state incompetent for chromatin recruitment (**A**). At the metaphase to anaphase transition (**A–B**), *D. melanogaster* ORC is activated, possibly in part through IDR dephosphorylation, thereby driving the condensation of ORC onto the surface of chromosomes (**B**). The Orc1 IDR likely plays a key role in this event, both interacting with DNA and also participating in intermolecular IDR-IDR interactions that drive ORC enrichment on chromosomes. The spatial patterning of ORC is developmentally regulated and appears correlated with the establishment of chromatin territories and TADs. Before cellularization (top, 'Pre-cellularization cycles'), the multi-nucleate fly embryo undergoes synchronous divisions that lack a G1 phase. At this stage, ORC shows homogenous chromosome binding. Conversely, ORC is non-uniformly distributed in differentiated cells (bottom, 'Lineage-dependent cycles') where replication domain boundaries coincide with underlying features of chromatin architecture. Once ORC is bound to chromosomes, Cdc6 and Cdt1 are co-recruited through direct interactions with DNA and through inter-initiator IDR-IDR interactions (**C**). When the full suite of helicase loading factors are present, the Pre-RC forms and Mcm2-7 loading commences (**D**). Loading is terminated by the phosphorylation of initiator IDRs which displace them from chromatin and inhibit re-association (**E**).

DOI: https://doi.org/10.7554/eLife.48562.019

physiological salt and protein concentrations. Moreover, the selective partitioning of Mcm2-7 into initiator/DNA phases is seen only when all three initiation components (ORC, Cdc6, and Cdt1) and ATP are present. This stringent requirement for the intact OCC and nucleotide is consistent with what is known to be the minimal set of factors necessary to catalyze the loading of Mcm2-7 complexes onto DNA. Since the pre-RC catalyzes multiple rounds of Mcm2-7 deposition through the reiterative binding and release of individual initiation factors (*Ticau et al., 2015*), the 'wetting' of a swath of DNA loci by a multitude of initiation factors would help keep those factors that cycle on and off of ORC at a relatively high concentration to promote successive Mcm2-7 loading cycles. Such behavior may be particularly useful in the early stages of embryogenesis and in pluripotent stem cells, where cycling times are short and helicase loading occurs rapidly (*Matson et al., 2017*).

Initiator IDRs may serve a role beyond simply increasing the efficiency of chromatin association and helicase loading. It has long been recognized that initiators are under stringent cellular control by CDKs (reviewed in *Parker et al., 2017*). It has also been established that a majority of CDK sites map to unstructured regions of Orc1, Cdc6, and Cdt1; however, how phosphorylation at these sites might regulate initiator function has remained unknown. Here, we demonstrate that the formation of initiator condensates is disrupted by CDK action (*Figure 6*). The inability of the phosphorylated initiator proteins to phase separate with DNA provides a ready explanation for how post-translational modifications unique to metazoan ORC could lead these factors to dissociate from chromatin late in the cell cycle and then associate again either in anaphase or G1 when the CDK levels are low (*Findeisen et al., 1999*; *Lee et al., 2012*; *Remus et al., 2005*) (). Budding yeast ORC is also negatively regulated by CDK-dependent phosphorylation (*Nguyen et al., 2001*), although this regulation likely occurs through a distinct mechanism, as we did not observe phase separation of *Sc*ORC (*Figure 3—figure supplement 1E*). It is notable that in metazoans Cdt1 inactivation requires the further degradation of the protein through a PCNA-dependent ubiquitination pathway (*Arias and Walter, 2006*); why this extra level of stringency has evolved remains to be determined. How phosphorylation interferes with phase separation is unknown and will require a more sophisticated understanding of the biophysical interactions underpinning the condensation reaction. Given the overall clustering of positively charged residues in the IDRs, and the DNA dependence of the condensation, charge neutralization of the domain and/or charge repulsion seem likely to play a critical role.

Beyond cell-cycle regulation, the IDR-dependent recruitment and co-association of ORC molecules with chromatin could also help account for more complex forms of initiator behavior and spatial patterning seen in vivo, such as the non-uniform organization of both replication origins and ORC binding sites across chromatin (*Cayrou et al., 2011*; *Miotto et al., 2016*; *Petryk et al., 2016*; *Vaughn et al., 1990*). We and others have observed that the distribution of ORC changes throughout the cell cycle, often in a species and cell-type specific manner (*Baldinger and Gossen, 2009*; *Kara et al., 2015*; *McNairn et al., 2005*; *Méndez et al., 2002*; *Natale, 2000*; *Sun et al., 2002*), and seems coincident with underlying changes in chromatin topology. For example, in early *D. melanogaster* embryos, in which the cell cycle proceeds directly from mitosis to S-phase (skipping G1), ORC appears to homogeneously coat mitotic chromosomes (*Figure 5* and *Figure 5—video 1– 2*). Human Orc1 shows a similar pattern of binding during mitosis in U2OS cells and, after mitotic exit, recruits other ORC subunits to chromatin (*Kara et al., 2015*). Mirroring the spatial patterning seen for human and *D. melanogaster* ORC in mitosis, mitotic chromosomes in metazoans are homogeneously organized and lack chromatin domains (*Naumova et al., 2013*; *Oomen and Dekker, 2017*). Conversely, ORC shows a non-uniform, punctate distribution in interphase in many differentiated cell types of metazoan species (*Austin et al., 1999*; *Kara et al., 2015*; *Lidonnici et al., 2004*; *Prasanth et al., 2010*; *Shen et al., 2010*). The switch in cellular ORC patterning between early embryonic and differentiated states appears coincident with the establishment of large-scale chromatin domains and topologically-associated domains (TADs) (*Hug et al., 2017*; *Li et al., 2014*), chromosomal regions that provide a basic level of chromatin organization and that also appear to possess liquid-phase like properties (*Nuebler et al., 2018*). Notably, replication domains appear to substantially overlap with TADs in multiple eukaryotic systems (*Petryk et al., 2016*; *Pope et al., 2014*). Similar to the role observed for the Orc1 IDR in binding mitotic chromosomes, it seems plausible that this region could help regulate large-scale spatial patterning of ORC in the nucleus. In such instances, the self-assembling properties of the Orc1 IDR would be used to reinforce the formation and maintenance of chromatin domains with other resident factors (such as HP1). Further

studies will be needed to explore whether initiator self-assembly on chromatin is coordinated with or helps to underpin higher-level replication organization ().

From a parallel perspective, it is interesting to note that the visualization of both newly-replicated DNA (*Kennedy et al., 2000*; *Xiang et al., 2018*) and proteins present at the replication fork during S-phase (*Chagin et al., 2016*) often reveals a non-uniform organization within the nucleus, a feature that has led to the proposal that replisomes may be grouped into 'factories' (*Cook, 1999*). While speculative, the formation and regulation of these replication foci could be aided by self-assembly mechanisms similar to those proposed here for initiation. Preliminary analyses of *D. melanogaster* replication proteins reveals that there is a high prevalence of long IDRs (data not shown). Proteins required at each stage of the replication reaction, from initiation to elongation, contain disordered domains, including Mcm10 and Chiffon (the fly homolog of Dbf4), as well as multiple cyclin proteins, topoisomerases, the clamp loader subunit Rfc1 and various polymerase components. Some of these IDRs have sequence characteristics similar to initiators, while others are clearly distinct. Given the selectivity we observe for initiator condensates (*Figure 4*), it is possible that the phase formed by Pre-RC machineries may represent an evolving structure that, through the fluid exchange of its resident components facilitated by IDR-IDR interactions, drives replication forward, culminating in the spatial co-localization of multiple active forks within a nuclear zone.

## Molecular mechanism of initiator phase separation

Phase transitions in biology are driven by multivalent interactions (*Li et al., 2012*). For protein LLPS, multivalency is achieved in two non-mutually exclusive ways: 1) by the linear and repetitive arrangement of folded interaction modules, and 2) through extended regions of intrinsic disorder that contain short linear interaction motifs (*Boeynaems et al., 2018*). Replication initiation factors fall within the latter of the two classes.

A major family of IDRs that mediate phase separation are the so-called low-complexity domain (LCD) proteins, which include factors such as FUS, Laf1, Ddx4 and hnRNPA1. Studies investigating the biophysical mechanisms of phase separation in these and other proteins have suggested a role for π-π and π-cation contacts (*Vernon et al., 2018*). Aromatic residues are major mediators of these type of interactions and have demonstrated functionality in LCD condensates (*Chong et al., 2018b*; *Jiang et al., 2015*; *Kato et al., 2012*; *Lin et al., 2017*; *Nott et al., 2015*; *Qamar et al., 2018*; *Wang et al., 2018*). Other sequences are also able to participate in π-π/π-cation bonding pairs, such as repetitive RG/RGG motifs (*Chong et al., 2018a*; *Elbaum-Garfinkle et al., 2015*). Even for non-LCD IDRs, such as the Nephrin intracellular domain (NICD), aromatic residue-mediated π-π contacts are of primary importance in facilitating phase separation (*Pak et al., 2016*). Given these data, we were surprised to find that both aromatic residues (<3% for Orc1, Cdc6, and Cdt1) and glycine (<4% for Orc1, Cdc6, and Cdt1) are substantially underrepresented in initiator IDR sequences compared to other phase separating LCD proteins (*Figure 5—figure supplement 1E*). Initiator IDRs are also devoid of simple repetitive motifs.

Outside of π-π/π-cation interactions, electrostatics can be another major driver of protein phase separation. This force is particularly relevant in systems that undergo complex coacervation with nucleic acids (*Aumiller and Keating, 2016*), such as seen with the tau protein and RNA (*Wegmann et al., 2018*; *Zhang et al., 2017*). The process of complex coacervation through the utilization of RNA as a counterion scaffold is mechanistically analogous to how we believe the replication initiators behave in the presence of DNA. Lines of evidence supporting this model include the demonstration that: 1) all *D. melanogaster* initiator IDRs have a pI >9 (*Table 1*), 2) initiator condensation is salt-sensitive (*Figure 2*) and each initiator has relatively high fraction of charged residues (between 32–35%) (*Figure 5—figure supplement 1*), and 3) although similar in overall amino acid composition, the Orc1 IDR (pI = 10.1) but not the Orc2 IDR (pI = 6.0) is necessary for ORC phase separation (*Figure 3*). We envision that initiator IDR interactions with DNA overcome interchain electrostatic repulsion between proximally positioned initiator IDRs, thereby promoting intermolecular interactions between the IDRs. Although at present we possess a minimal understanding of the biophysics of initiator LLPS, the relatively equal representation of hydrophobic, hydrophilic, and charged amino acids within initiator IDRs suggests that intermolecular IDR-IDR interactions may be of multiple types (*Figure 5—figure supplement 1*).

## Conservation of initiator IDRs across eukaryotes

The IDRs of *D. melanogaster* Orc1, Cdc6, and Cdt1 share a similar sequence composition but no identifiable sequence identity. This profile has led us to conclude that the general sequence features of these IDRs are sufficient to facilitate DNA-dependent phase separation. It is notable, however, that the Orc1 IDR amino acid sequences are highly conserved across the *Drosophila* genus (representing over 40 M years of evolution; *Drosophila 12 Genomes Consortium et al., 2007*; *Figure 3— figure supplement 2*), suggesting that there may be presently unrecognized sequence-specific functional information.

All metazoan homologs of Orc1, Cdc6, and Cdt1 that we have analyzed contain a predicted N-terminal IDR longer than one-hundred amino acids (*Table 1*, including human, mouse, *D. rerio, X. laevis*, and *C. elegans*). Interestingly, for all species Orc1 contains the longest IDR and Cdc6 the shortest. While suggestive, an IDR alone does not unequivocally determine whether a protein will phase separate. Further classification of eighteen homologous metazoan sequences demonstrate that they contain sequence characteristics similar to *D. melanogaster* initiator proteins: all possess a basic pI (calculated pIs range from 9.1 to 11.1), a low content of aromatic and glycine residues, and a relatively high fraction of charged residues (ranging from 0.22 to 0.39). Based on our current limited understanding of initiator phase separation, these data suggest that the ability of initiators to undergo DNA-dependent LLPS is likely broadly conserved across metazoa. In line with this idea, we show that human Cdt1, like *Dm*Cdt1, forms condensates in a DNA-dependent manner (*Figure 2— figure supplement 1G*). A recent report demonstrating an essential function for the phosphorylation-regulated conversion of intramolecular IDR interactions in human Orc1 to intermolecular interactions between ORC and Cdc6 IDRs (*Hossain et al., 2019*) is similarly supportive with such a proposal (*Figure 6*). We anticipate that these interactions, possibly augmented by other transient IDR-IDR interactions forming in both *cis* and *trans*, facilitate some form of condensation or localized clustering of initiation factors for the functional assembly of pre-RCs. In instances where Mcm2-7 loading has been reported using purified metazoan proteins (e.g., human; *Wu et al., 2014*), it is tempting to speculate that this may be occurring in such environments.

In contrast to the metazoan initiators, our sequence analysis predicts that the *S. cerevisiae* initiation factors are likely to be incapable of DNA-dependent phase separation (*Table 1*). Indeed, we found *Sc*Cdt1, which lacks an N-terminal IDR altogether, unable to either bind DNA or undergo phase separation (*Figure 2—figure supplement 1*). Additionally, the IDRs of *Sc*Orc1 and *Sc*Cdc6 are relatively short (143 and 31 amino acids, respectively) and have an acidic as opposed to a basic pI (pI = 4.7 and 6.2, respectively); *Sc*ORC also showed no evidence of phase separation (*Figure 3— figure supplement 1E*). For fission yeast, the predictions from our analysis are less reliable. *S. pombe* Cdt1 does not have an N-terminal IDR but homologs of Orc1 and Cdc6 each have an approximately one-hundred amino acid N-terminal IDR with a basic pI.

## Intersection with emerging roles of cellular phase separation

The present work provides strong evidence that DNA replication, a cellular pathway vital to cell proliferation, is impacted by phase separation-promoting elements. These observations in turn provide a provocative link between the replication machinery and their function in at least two other cellular structures recently predicted to be impacted by phase separation. One is heterochromatin: recent work has demonstrated that Heterochromatin Protein 1 (HP1), a major cellular organizer of heterochromatin, undergoes phase separation in vitro and in vivo (*Larson et al., 2017*; *Strom et al., 2017*). Interestingly, there is a conserved linkage between the formation of HP1-dependent heterochromatic domains and ORC binding, with the localization of HP1 and ORC to chromatin being interdependent (*Huang et al., 1998*; *Pak et al., 1997*; *Prasanth et al., 2010*; *Prasanth et al., 2004*). Notably, Orc1 targeting to heterochromatin relies on the Orc1 N-terminal IDR in both human and *D. melanogaster* model systems (*Lidonnici et al., 2004*; *Pak et al., 1997*). It is tempting to speculate that the formation of heterochromatic domains may involve a condensed phase that forms by the coordinated action of ORC and HP1 (and likely many other proteins), and that this activity may have the dual purpose of excluding transcriptional machineries to maintain a silenced state while enriching ORC in an otherwise inaccessible compartment that must nonetheless be replicated. We anticipate that any mechanism(s) underlying the formation and maintenance of heterochromatic domains may be developmentally regulated, since HP1 domains are clearly visible during interphase

in the early embryo (*Strom et al., 2017*), whereas the association of ORC with chromatin appears restricted to mitosis during the same developmental stage (*Figure 5*). Beyond HP1, ORC phase separation may have a direct role in chromatin compaction, which could underlie reports that ORC plays a role in mitotic chromosome condensation (*Pflumm and Botchan, 2001*; *Prasanth et al., 2004*).

A second condensate connection with ORC may involve centrosomes. It has recently emerged that centrosomes form by the phase separation of scaffolding factors, known as the pericentriolar material (PCM). A major PCM scaffolding factor in *C. elegans*, SPD-5, undergoes spontaneous self-assembly into spherical condensates that selectively recruit centrosome client proteins necessary for the nucleation of microtubules (*Woodruff et al., 2017*; *Woodruff et al., 2015*). Notably, multiple subunits of ORC, including both Orc1 and Cdc6, are targeted to centrosomes where they regulate centrosome copy number (*Hemerly et al., 2009*; *Kim et al., 2015*; *Prasanth et al., 2004*). We speculate that the sequence features which enable initiator phase separation on DNA (e.g. their N-terminal IDRs) may permit their selective partitioning and enrichment on centrosomes. From a mechanistic standpoint, it is interesting to note that the scaffolding protein SPD-5 and its homologs in other metazoans bear a high net negative charge (*Dm*Centrosomin is 30% charged with pI = 5.8), suggesting that initiator recruitment to centrosomes may proceed through a mechanism similar to their condensation on chromatin, that is, by complex coacervation through charge-charge interactions with a poly-anionic scaffold (in this case, protein).

In conclusion, we have demonstrated that replication initiators contain a novel class of IDRs that facilitate phase separation and affect each stage of the initiator functional cycle. This work establishes a new paradigm for understanding metazoan initiation mechanisms and their regulation by CDK activity, and how initiation can be organized at the mesoscale. These findings in turn open up numerous new avenues of cross-disciplinary investigation into how the replication initiation machineries interface with other cellular pathways, such as those involved in chromatin maintenance and establishment.

# Materials and methods

**Key resources table**

| Reagent type (species) or resource | Designation | Source or reference | Identifiers | Additional information |
|---|---|---|---|---|
| Recombinant DNA reagent | 2Cc-T | QB3 Macrolab (UC Berkeley) | RRID:Addgene_37237 | Ligation independent cloning (LIC); *E. coli* expression vector |
| Recombinant DNA reagent | 1GFP | QB3 Macrolab (UC Berkeley) | RRID:Addgene_29663 | LIC cloning; *E. coli* expression vector |
| Recombinant DNA reagent | 1b | QB3 Macrolab (UC Berkeley) | RRID:Addgene_29653 | LIC cloning; *E. coli* expression vector |
| Recombinant DNA reagent | pFastbac1 | ThermoFisher | | Insect cell expression vector |
| Recombinant DNA reagent | 438A | QB3 Macrolab (UC Berkeley) | RRID:Addgene_55218 | LIC cloning; insect cell expression vector |
| Recombinant DNA reagent | 438B | QB3 Macrolab (UC Berkeley) | RRID:Addgene_55219 | LIC cloning; insect cell expression vector |
| Recombinant DNA reagent | 4C | QB3 Macrolab (UC Berkeley) | RRID:Addgene_30116 | LIC cloning; insect cell expression vector |
| Recombinant DNA reagent | pattB | | | Fragments inserted into MCS by restriction enzyme cloning; transgene vector |
| Recombinant DNA reagent | pCopia-LIC | This paper | | LIC cloning; *D. melanogaster* cell culture expression vector |

*Continued on next page*

*Continued*

| Reagent type (species) or resource | Designation | Source or reference | Identifiers | Additional information |
|---|---|---|---|---|
| Peptide, recombinant protein | TEV | QB3 Macrolab (UC Berkeley) | | Used at 1/10 (weight/weight) TEV/substrate ratio |
| Peptide, recombinant protein | Flag peptide | Sigma-Aldrich | | |
| Strain, strain background (*Escherichia coli*) | Rosetta 2(DE3)pLysS | QB3 Macrolab (UC Berkeley) | | Chemically competent cells |
| Strain, strain background (*Escherichia coli*) | DH10bac | QB3 Macrolab (UC Berkeley) | | Chemically competent cells |
| Cell line (*D. melanogaster*) | S2 | UC Berkeley Cell Culture Facility | | |
| Cell line (*Spodoptera frugiperda*) | Sf9 | UC Berkeley Cell Culture Facility | | |
| Cell line (*Trichoplusia ni*) | High5 | This paper; UC Berkeley Cell Culture Facility | | |
| Commercial assay or kit | ANTI-FLAG M2 Affinity Agarose Gel | Sigma-Aldrich | A2220 (RRID:AB_10063035) | 0.5 mL resin per 1 L expression |
| Commercial assay or kit | Amylose Resin | NEB | E8021L | 5 mL of resin; column format |
| Commercial assay or kit | High Capacity Strepatavidin Agarose | ThermoFisher Scientific | 20357 | |
| Commercial assay or kit | Effectene | Qiagen | 301425 | |
| Antibody | Anti-Cdt1 affinity purified antibody (rabbit polyclonal) | M. Botchan | | 1/1,000 dilution |
| Antibody | IRDye800 CW Donkey anti-Rabbit (donkey polyclonal) | LI-COR | 926–32213 (RRID:AB_621848) | 1/10,000 dilution |
| Antibody | Anti-FLAG (rabbit monoclonal) | Sigma-Aldrich | F7425 | 1/1,000 |

## Cloning, expression, and purification of DmCdt1

The coding sequence for full-length *D. melanogaster* Cdt1 was cloned into the QB3 Macrolab vector 2Cc-T for a tobacco etch virus (TEV) protease-cleavable C-terminal hexa-histidine (His6)-maltose binding protein (MBP) tag and into vector 1GFP for an N-terminal His6-eGFP tag. A construct lacking the N-terminal IDR (amino acids 298–743), Cdt1$^{\Delta IDR}$, was cloned into QB3 Macrolab vector 1b for a TEV-cleavable N-terminal His6 tag. All DmCdt1 constructs were expressed in overnight cultures at 16°C from Rosetta 2(DE3)pLysS (QB3 Macrolab) after 1 mM IPTG induction. Cells were harvested by centrifugation and cell pellets frozen at −80°C until further processing.

A cell pellet from 1 L of DmCdt1 expressing cells was resuspended in 40 mL of Lysis Buffer (20 mM Tris pH 7.5, 500 mM NaCl, 30 mM Imidazole, 10% glycerol, 200 μM PMSF, 1x cOmplete EDTA-free Protease Inhibitor Cocktail (Sigma-Aldrich), 1 mM BME and 0.1 mg/mL lysozyme) and lysed by sonication. The lysate was clarified by centrifugation at 30,000 xg for 1 hr, filtered through an aPES 0.45 μm bottle-top filter unit (Nalgene Rapid-Flow, ThermoFisher), and then passed over a 5 mL HisTrap HP column (GE Healthcare). The column was washed with 10 column volumes (CV) of Nickel Wash Buffer (20 mM Tris pH 7.5, 500 mM NaCl, 30 mM Imidazole, 10% glycerol, 200 μM PMSF, 1 mM BME) and finally eluted with 15 mL of Nickel Elution Buffer (20 mM Tris pH 7.5, 150 mM NaCl, 500 mM Imidazole, 10% glycerol, 1 mM BME). The protein was further purified by heparin affinity chromatography (HiTrap Heparin HP, GE Healthcare), eluting with a linear gradient of increasing salt from 150 mM - 1 M NaCl; full-length Cdt1 elutes at approximately [NaCl]=600 mM while

degradation products elute earlier. Fractions containing full-length Cdt1 were then pooled and, for full-length and Cdt1$^{\Delta IDR}$, but not eGFP-Cdt1, the tag was cleaved with TEV protease at a 1:10 ratio of TEV:protein in an overnight incubation at 4°C. An additional nickel affinity step was then used to remove TEV, uncleaved protein, and the free tag from purified Cdt1. Finally, the purified sample was concentrated to 2 mL and loaded onto a HiPrep 16/60 Sephacryl S-300 HR column (GE Healthcare) equilibrated and run in Sizing Buffer (50 mM HEPES pH 7.5, 300 mM KGlutamate, 10% glycerol, 1 mM BME). Peak fractions were pooled, concentrated in a 10K Amicon Ultra-15 concentrator (Millipore), flash frozen in liquid nitrogen, and stored at −80°C. All DmCdt1 constructs were purified with the same procedure, except for eGFP-Cdt1 for which the TEV cleavage step was omitted.

## Cloning, expression, and purification of ScCdt1

The coding sequence for full-length S. cerevisiae Cdt1 was cloned into the QB3 Macrolab vector 1b for expression with a TEV-cleavable N-terminal His6-tag. ScCdt1 was expressed in bacteria as previously described for DmCdt1. Cells from a 1 L expression were resuspended in 40 mL of Lysis Buffer (50 mM Tris pH 7.5, 300 mM KCl, 10% glycerol, 30 mM Imidazole, 200 µM PMSF, 1x cOmplete EDTA-free Protease Inhibitor Cocktail (Sigma-Aldrich), 1 mM BME and 0.1 mg/mL lysozyme) and lysed by sanitation. The lysate was clarified by centrifugation at 30,000 xg for 1 hr and subsequently filtered through an aPES 0.45 µm bottle-top filter unit. The clarified lysate was loaded onto a 5 mL HisTrap HP column and washed with 10 CV of Nickel Wash Buffer (50 mM Tris pH 7.5, 300 mM KCl, 10% glycerol, 30 mM Imidazole, 200 µM PMSF, 1 mM BME) and eluted with 3 CV of Nickel Elution Buffer (50 mM Tris pH 7.5, 300 mM KCl, 10% glycerol, 250 mM Imidazole, 1 mM BME). The His6 tag was then removed with an overnight incubation with TEV protease at a 1:10 (w:w) ratio of TEV:protein. After cleavage, cleaved ScCdt1 was isolated from TEV, the cleaved tag, and uncleared protein by an additional nickel purification step. Finally, the protein was purified by size exclusion chromatography over a HiPrep 16/60 Sephacryl S-300 HR column (GE Healthcare) equilibrated and run in Sizing Buffer (50 mM HEPES pH 7.5, 300 mM KGlutamate, 10% glycerol, 1 mM BME). Peak fractions were pooled, concentrated in a 10K Amicon Ultra-15 concentrator (Millipore), flash frozen in liquid nitrogen, and stored at −80°C.

## Cloning, expression, and purification of human Cdt1

The coding sequence for full-length human Cdt1 was cloned into the QB3 Macrolab vector 4C for expression with a tobacco etch virus (TEV)-cleavable C-terminal hexa-histidine (His6)-maltose binding protein (MBP) tag. This vector was transformed into DH10bac cells for generation of bacmid DNA that was transfected (Cellfectin II) and amplified in Sf9 cells. Virus was amplified twice in Sf9 cells and protein expressed from 2 L of High5 cells infected with high-titer virus. Human Cdt1 was purified from High5 cells according to the method described above for DmCdt1 except that lysozyme was omitted from the Lysis Buffer.

*Cloning, expression, and purification of DmORC.* All ORC constructs were expressed in High5 insect cells from baculoviruses. Baculoviruses were generated by Cellfectin II (ThermoFisher) transfection of Sf9 cells with DH10bac-derived bacmid DNA that, in all cases, encoded a TEV-cleavable His6 and MBP tag on Orc1 and Orc4, respectively. Full-length ORC was generated by co-infection of High5 cells with two baculoviruses, one encoding Orc1-5 and the second encoding Orc6. ORC complexes were also produced with deletions in the N-terminal IDR of Orc1 (*ORC$^{1\Delta IDR}$*) and Orc2 (*ORC2$^{\Delta IDR}$*). ORC$^{1\Delta IDR}$ was generated by co-infection of High5 cells with three baculoviruses, one encoding Orc2-5, a second encoding Orc6, and a final virus from which Orc1$^{\Delta IDR}$ was expressed. To generate the deletion, Orc1 was cloned into Macrolab vector 438B and an internal deletion (deletion of amino acids 248–549) generated by around-the-horn mutagenesis. Similarly, *ORC$^{2\Delta IDR}$* was generated by co-infection of High5 cells with three baculoviruses, one encoding both Orc1 and Orc3-5, a second encoding Orc6, and a final virus from which Orc2$^{\Delta IDR}$ was expressed. To generate the deletion, an Orc2 N-terminal deletion (amino acid 270–618) was cloned into Macrolab vector 438A. Baculoviruses were amplified for two rounds prior to infection of High5 cells for protein expression.

All ORC constructs were purified according to the same procedure. The cell pellet from 2 L of cells was resuspended in 80 mL of Lysis Buffer (50 mM Tris pH 7.5, 300 mM KCl, 50 mM Imidazole, 10% glycerol, 200 µM PMSF, 1x cOmplete EDTA-free Protease Inhibitor Cocktail (Sigma-Aldrich), 1 mM BME) and lysed by sonication. The lysate was then clarified by centrifugation at 30,000 xg for 1

hr and the supernatant filtered through an aPES 0.45 µm bottle-top filter unit. Subsequently, the lysate was subjected to a 20% ammonium sulfate precipitation for 30 min at 4°C and then centrifuged again for 1 hr at 30,000 xg. The supernatant was then passed over a 5 mL HisTrap HP column (GE Healthcare) and washed with 10 CV of Nickel Wash Buffer (50 mM Tris pH 7.5, 300 mM KCl, 50 mM Imidazole, 10% glycerol, 200 µM PMSF, 1 mM BME) before protein elution with a 6 CV linear gradient from 50 to 250 mM Imidazole. The sample was then further purified by a second affinity step over an 8 mL amylose column (NEB), where it was washed with 3 CV of Amylose Wash Buffer (50 mM Tris pH 7.5, 300 mM KCl, 10% glycerol, 1 mM BME) and eluted with 2 CV of Amylose Elution Buffer (Amylose Wash Buffer supplemented to 20 mM maltose). The affinity tags from Orc1 and Orc4 were then removed by adding TEV at a 1:10 ratio and incubating overnight at 4°C. Finally, the sample was concentrated and then purified by size exclusion chromatography on a HiPrep 16/60 Sephacryl S-300 HR column (GE Healthcare) equilibrated and run in Sizing Buffer (50 mM HEPES pH 7.5, 300 mM KGlutamate, 10% glycerol, 1 mM BME). Peak fractions were pooled, concentrated in a 30K Amicon Ultra-15 concentrator (Millipore), flash frozen in liquid nitrogen, and stored at −80°C.

## Cloning, expression, and purification of DmCdc6

The coding sequence for full-length *D. melanogaster* Cdc6 and a construct lacking the N-terminal IDR (amino acids 231–662), Cdc6$^{\Delta IDR}$, was cloned into the QB3 Macrolab vector 4C for expression in Sf9 insect cells as an TEV-cleavable N-terminal His6-MBP fusion. To generate tagRFP-Cdc6, the coding regions for tagRFP and Cdc6 were amplified and cloned by Gibson Assembly into vector 4C to generate tagRFP-Cdc6 with a TEV-cleavable N-terminal His6-MBP tag. Expression vectors were transformed into DH10bac cells (QB3 Macrolab) for production of bacmid DNA that was subsequently transfected into Sf9 cells using Cellfectin II to generate virus. Baculoviruses were amplified for two rounds in Sf9 cells to generate the high-titer virus used for expression.

For expression of Cdc6, one liter of Sf9 cells in a shaker flask was infected with high-titer baculovirus for two days. Subsequently, the cells were harvested by centrifugation and stored at −80°C until downstream processing. To purify Cdc6, the 1 L of harvested cells was resuspended in 40 mL of Lysis Buffer (50 mM Tris pH 7.5, 300 mM KCl, 10% glycerol, 200 µM PMSF, 1x cOmplete EDTA-free Protease Inhibitor Cocktail (Sigma-Aldrich), 5 mM MgOAc, 10 µM ATP, 1 mM BME) and lysed by sonication. The lysate was clarified by centrifugation at 30,000 xg for 1 hr, filtered through an aPES 0.45 µm bottle-top filter unit (Nalgene Rapid-Flow, ThermoFisher), and then passed over an 5–8 mL amylose column (NEB). The bound protein was washed with ten CV of Amylose Wash Buffer (50 mM Tris pH 7.5, 300 mM KCl, 10% glycerol, 200 µM PMSF, 5 mM MgOAc, 10 µM ATP, 1 mM BME) and eluted with 2 CV of Amylose Elution Buffer (Amylose Wash Buffer supplemented to 20 mM maltose). Peak fractions were pooled and TEV protease added at a 1:10 ratio of TEV to protein, then incubated overnight at 4°C. The cleaved protein was subsequently purified by heparin affinity chromatography (HiTrap Heparin HP, GE Healthcare), eluting with a linear gradient of increasing salt from 100 mM - 1 M KCl. Finally, the peak fractions from heparin were concentrated and purified by size exclusion chromatography over a Hiprep 16/60 Sephacry S-300 HR column equilibrated in Sizing Buffer supplemented with 5 mM MgOAc and 10 µM ATP. Peak fractions were pooled, concentrated in a 10K Amicon Ultra-15 concentrator (Millipore), flash frozen in liquid nitrogen, and stored at −80°C.

## Cloning, expression, and purification of HsFUS

Full-length human FUS was cloned into QB3 Macrolab vector 438A with an N-terminal His6-eGFP tag with intervening TEV consensus sequence. Bacmid DNA was generated by transformation of the expression vector into DH10bac cells and the resulting bacmid transfected into Sf9 cells for generation of baculovirus which was amplified twice to generate high-titer virus. High5 cells were infected with virus for two days and then harvested and frozen for later purification. One liter of infected cells was resuspended with 40 mL of Lysis Buffer (50 mM Tris pH 7.5, 1 M KCl, 30 mM Imidazole, 10% glycerol, 200 µM PMSF, 1x cOmplete EDTA-free Protease Inhibitor Cocktail (Sigma-Aldrich)), 1 mM BME) and lysed by sonication. The lysate was then centrifuged at 30,000 xg for 1 hr and the clarified lysate passed through an aPES 0.45 µm bottle-top filter unit. The protein was then passed over a 5 mL HisTrap HP column (GE Healthcare), washed with 10 CV of Nickel Wash Buffer (50 mM Tris pH 7.5, 1 M KCl, 30 mM Imidazole, 10% glycerol, 200 µM PMSF, 1 mM BME), and eluted with 3 CV of

Nickel Elution Buffer (50 mM Tris pH 7.5, 1 M KCl, 300 mM Imidazole, 10% glycerol, 1 mM BME). The protein was then concentrated with a 10K Amicon Ultra-15 concentrator (Millipore) and diluted into Storage Buffer (50 mM Tris pH 7.5, 1 M KCl, 150 mM Imidazole, 10% glycerol, 1 mM BME) prior to aliquoting, snap freezing in liquid nitrogen, and storing at −80°C.

## Cloning, expression, and purification of DmMcm2-7

The DmMcm2-7 complex was expressed and purified from High5 insect cells co-infected with a single virus for each subunit. The coding region for MCM3-7 were cloned into pFastbac1 (Thermo-Fisher). MCM3 was cloned with an N-terminal FLAG tag (DYKDDDDK, ThermoFisher). The coding sequence for MCM2 was cloned into QB3 Macrolab vector 4C to generate an N-terminal His6-MBP fusion. Each vector was transformed into DH10bac cells to generate bacmid DNA that was transfected (Cellfecin II) into Sf9 cells and amplified twice to generate high-titer virus prior to infection of High5 cells for protein expression. Infected High5 cells were grown for 2 days and then the cells were pelleted and froze at −80°C for later processing. The cells were resuspend in Lysis Buffer (25 mM HEPES pH 7.5, 15 mM KCl, 10% glycerol, 0.08% Tween-20, 2 mM EDTA, 2 mM EGTA, 800 μM PMSF, 1x cOmplete EDTA-free Protease Inhibitor Cocktail (Sigma-Aldrich)) and lysed by dounce homogenization. After lysis, the lysate was incubate for 10 min on ice and the solution adjusted to [KCl]=100 mM. The lysate was clarified by centrifugation for 15 min at 30,000 xg and was then incubated for two hours with ANTI-FLAG M2 Affinity Agarose Gel (Sigma-Aldrich, A2220). The resin was then washed with 40 CV of Flag Wash Buffer (25 mM HEPES pH 7.5, 200 mM KAcetate, 10% glycerol, 2 mM EDTA, 2 mM EGTA, 800 μM PMSF) and eluted by competitive elution in Flag Wash Buffer supplemented to 200 ug/mL Flag peptide and 250 ug/mL human insulin as a carrier (Sigma-Aldrich). The protein was then passed over a 2 mL amylose column (NEB) and washed with 3 CV of Flag Wash Buffer. The protein was eluted with 3 CV Flag Wash Buffer supplemented to 20 mM maltose. The His6-MBP tag was removed with an overnight digestion with TEV protease (QB3 Macrolab). Finally, the sample was purified by size exclusion chromatography over a Superose6 5/150 GL equilibrated and run in Protein Buffer (50 mM HEPES pH 7.5, 300 mM KGlutamate, 10% glycerol, 1 mM BME). Concentrated Mcm2-7 was aliquoted, flash frozen in liquid nitrogen, and then stored at 80°C.

## Cloning, expression, and purification of DmCDK1/CycA complex

The DmCDK1/CycA complex was expressed in High5 insect cells co-infected with two viruses, one encoding CDK1 and the other CycA. Full-length CDK1 was cloned into QB3 Macrolab vector 4C for expression with a TEV-cleavable His6-MBP tag and full-length CyclinA was cloned into QB3 Macrolab vector 4B for expression with a TEV-cleavable His6 tag. Viruses were generated in Sf9 cells by Cellfectin II transfection of bacmid DNA (derived from DH10bac cells) which were then amplified for two rounds in Sf9 cells.

Two liters of cells co-infected with CDK1 and CycA were resuspended in 80 mL of Lysis Buffer (50 mM HEPES pH 7.5, 300 mM KCl, 30 mM Imidazole, 10% glycerol, 200 μM PMSF, 1x cOmplete EDTA-free Protease Inhibitor Cocktail (Sigma-Aldrich), 1 mM BME) and lysed by sonication. The lysate was then clarified by centrifugation at 30,000 xg for 1 hr and the supernatant filtered through an aPES 0.45 μm bottle-top filter unit. The supernatant was then passed over a 5 mL HisTrap HP column (GE Healthcare), washed with ten CV of Nickel Wash Buffer (50 mM HEPES pH 7.5, 300 mM KCl, 30 mM Imidazole, 10% glycerol, 200 μM PMSF, 1 mM BME), and finally eluted with a 10 CV linear gradient from 30 to 500 mM Imidazole. Peak fractions were pooled and passed over an 8 mL amylose column (NEB), washed with 5 CV of Amylose Wash Buffer (50 mM HEPES pH 7.5, 300 mM KCl, 10% glycerol, 1 mM BME), and eluted with 3 CV of Amylose Elution Buffer (Amylose Wash Buffer supplemented to 20 mM maltose). The affinity tags were then removed with an overnight digestion at 4°C with TEV added at a 1:10 (w:w) ratio. After an additional nickel affinity step to remove uncleaved protein, CDK1/CycA was concentrated and the holocomplex separated from free CDK1 by size exclusion chromatography on a Superose 6 Increase 10/300 GL column (GE Healthcare). Peak fractions were analyzed by SDS-PAGE analysis and fractions where CDK1 and CycA co-eluted were pooled, concentrated with a 10K Amicon Ultra-15 concentrator (Millipore), flash frozen in liquid nitrogen, and stored at −80°C.

## DNA pull-down assays

A random oligonucleotide of sixty basepairs was produced and annealed with a complementary oligonucleotide in Annealing Buffer (50 mM HEPES pH 7.5, 50 mM KCl). This oligonucleotide is referred to throughout the paper as 'dsDNA' and has sequence:

5′-GAAGCTAGACTTAGGTGTCATATTGAACCTACTATGCCGAACTAGTTACGAGCTATAAAC −3′. For DNA pull down assays, dsDNA was produced with a 5′ biotin label (biotin-dsDNA, IDT). To generate DNA-coupled agarose beads, high capacity streptavidin agarose resin (Pierce) was first washed with Annealing Buffer and then 25 µM biotin-dsDNA was added to the beads and incubated for 1 hr at room temperature; control beads were generated by adding Annealing Buffer alone. Following coupling, the beads were washed three times with Annealing Buffer. To assay DNA binding, the beads were first washed three times with Assay Buffer (50 mM HEPES pH 7.5, 150 mM KGlutamate, 10% glycerol, 1 mM BME) and then 10 µM Cdt1 or Cdc6 was added to control and DNA coupled beads (Cdc6 binding was assessed in the presence and absence of 500 µM ATP). The beads were incubated with protein for 1 hr at room temperature and then washed three times with 10x volume of Assay Buffer before resuspending and boiling the beads in Laemmmli sample buffer and assessing bound proteins by SDS-PAGE analysis and Coomassie staining.

## Fluorescence polarization DNA-binding assays

Fluorescence polarization readings were taken on a BioTek Synergy using 384-well black bottom plates (Greiner) and detection of Cy5-labeled dsDNA (Cy3-dsDNA). Ten 15 uL reactions were prepared in Assay Buffer (50 mM HEPES pH 7.5, 150 mM KGlutamate, 10% glycerol, 1 mM BME) containing 50 nM Cy3-dsDNA and a serial dilution of Cdt1 from 1 µM to 0.5 nM. The reactions were incubated 10 min at room temperature and then polarization measured relative to a control. The mean and standard deviation of three independent experiments were plotted as a function of protein concentration and the data fit in Prism with a Hill equation to determine the dissociation constant.

## Electrophoretic mobility shift assay (EMSA)

EMSA gels were imaged on an Odyssey imaging system (LI-COR) through detection of an IRDye800 fluorescent tag appended to the 5′ end dsDNA (IRDye800-dsDNA, IDT). Ten 20 µL reactions were prepared in Assay Buffer (50 mM HEPES pH 7.5, 150 mM KGluatmate, 10% glycerol, 1 mM BME) containing 25 nM IRDye800-dsDNA and a serial dilution of Cdt1 from 3 µM to 0.2 nM. The reactions were incubated for 45 min and then 5 µL of each sample was run on a 5% native PAGE gel at 100 V for 1 hr.

## Determination of embryonic Cdt1 concentration

*D. melanogaster* embryos (Bloomington stock 32045) were collected at 2 hr intervals from age 2–16 hr, dechorionated in 100% bleach for 60 s, and washed extensively with $H_2O$. For each time-point, the number of dechorionated embryos was counted and then resuspended with 1 µL of 1x Laemmli sample buffer for every one embryo. Samples were next homogenized with a micro-pestle (Sigma, Z359947), heated for 5 mins at 95℃, and then clarified with a 3 min centrifugation at 13,000 xg before transferring the supernatant to a new tube. The lysate from each time-point was fractionated on a 4–20% Bio-Rad TGX gel (alongside a dilution series of recombinant Cdt1 as a standard) and then transferred to a nitrocellulose membrane. After blocking with 5% BSA, the membrane was probed with an affinity purified anti-*Dm*Cdt1 antibody (1/1,000, laboratory of M. Botchan) followed by incubation with a secondary IRDye800 CW Donkey anti-Rabbit antibody (1,/10,000, LI-COR, 926–32213). Blots were imaged with a LI-COR Odyssey imager. A standard curve was generated from the intensity values of the recombinant Cdt1 dilution series and the per-embryo Cdt1 concentration calculated based on the reported volume of a *D. melanogaster* embryo (**Markow et al., 2009**).

## Assaying liquid–liquid phase separation

Multiple methods were used to assay for phase separation. For the turbidity assays, 20 µM *Dm*Cdt1 was combined with 20 µM dsDNA in microcentrifuge tubes; as controls, samples were prepared that contained only *Dm*Cdt1 or dsDNA. After mixing, the solution became immediately turbid and the tubes were imaged on an Epson Perfection V700 scanner. We asked whether this was a reversible

process by adding concentrated KCl (4 M) to a final concentration of 400 mM to the Cdt1/DNA mixture and again imaging the sample. To determine the source of the turbidity, 7 µL of each sample was spotted on a slide, covered with a coverslip, and imaged on a wide-field microscope equipped with DIC optics at a magnification of 63x with an oil immersion objective. For assaying droplet formation by fluorescence microscopy, samples were prepared by mixing protein with Cy5 or Cy3-labeled dsDNA (Cy5- or Cy3-dsDNA) and incubating 2 min prior to spotting 7 µL on a glass slide and covering with a glass coverslip. Subsequently, the samples were imaged on a Zeiss LSM 710 inverted confocal microscope using a 63x oil immersion objective and appropriate filter sets. For multi-color fluorescent imaging, control samples were imaged to ensure no crosstalk was observed between channels. Multi-color droplet recruitment assays were completed by first preparing droplets with ORC and Cy5-dsDNA. Subsequently, fluorescently-tagged proteins were spiked into each reaction and imaged. Quantitation of protein recruitment to preformed ORC droplets was completed in FIJI (*Schindelin et al., 2012*). First, regions of interest (ROI) for ORC/Cy5-dsDNA droplets were generated with the auto-threshold function and then the signal intensity for the eGFP channel was measured within these regions, form which a mean and standard deviation was calculated. Finally, depletion assay samples were prepared by mixing protein with an equimolar amount of dsDNA and incubating the samples for 30 min at room temperature. After incubation, the samples were centrifuged for 10 min at 16,000 xg and the supernatant removed for separation by SDS-PAGE analysis and subsequent Coomassie or Silver staining; in all cases, a protein load control was also assessed that had not been centrifuged. Depletion experiments that assayed the effect of DNA length and sequence on initiator phase separation were completed at 2 µM protein concentrations and 0.03 mg/mL of the following generated-randomly dsDNA oligonucleotides:

| Name | Sequence |
| --- | --- |
| 15mer: | CACAGCGTACTCACA |
| 25mer: | CACAGACGCACCAGTTTACACTCAG |
| 50mer: | CATGCATACACGAGCTGCACAAACGAGAGTGCTTGAACTGGACCTCTAGT |
| 60mer: | GAAGCTAGACTTAGGTGTCATATTGAACCTACTATGCCGAACTAGTTACGAGCTATAAAC |
| 10% GC: | CATTTAATAATTTTGTAATAAAAATTAAGAAAATAATAATAATTATAAATACTATCGTAT |
| 25% GC: | AAATGTTTCTTACAATAAAACGATCAAGTACATTTTTATAAAAGGTGATAGAGATTTACG |
| 50% GC: | GATACTTGGGCTTGATCTCGCCCCGACACCTGCAAACCTCAACTGCCTTAGATTATATGG |
| 75% GC: | GGTGGTGTCGGGTCAGGGCGGCCCCGCGACCAGTCGTGTGCCTTCCCGAGCTCCGTCCGG |

## Disorder and complexity calculations

The per-residue disorder score was calculated using the online DISOPRED3 disorder prediction server (*Jones and Cozzetto, 2015*) and N-terminal IDRs characterized by the longest contiguous stretch of disordered residues (DISOPRED >0.5). For the comparison of initiator IDRs with *Hs*FUS, the isolated initiator N-terminal IDRs were compared to all residues of *Hs*FUS with a disorder prediction score >0.5 and heatmaps of the corresponding regions were generated in excel using conditional formatting. These same regions were used to classify the IDRs according to amino acid type (aromatic = F, W, Y; hydrophobic = A, G, I, L, M, P, V; hydrophilic = C, N, Q, S, T; charged = D, E, H, K, R). The local compositional complexity was calculated for each amino acid in a 20-residue sliding window and the average complexity score compared for predicted disordered and ordered regions of the protein (*Wootton and Federhen, 1993*). The data are presented as a box plot of individual residue scores, where the middle line of the box = the median value, the top line = the limit of the upper quartile, the bottom line = the limit of the lower quartile, and the individual points represent outliers to the upper and lower extremes of the data, which are indicated as lines coming off the box.

## MCM recruitment assays

Loading reactions were prepared that contained *Dm*ORC (125 nM), *Dm*Cdc6 (125 nM), *Dm*Cdt1 (125 nM), *Dm*Mcm2-7 (250 nM), ATP (1 mM), MgOAc (4 mM), and plasmid (pBluescript) DNA (5 nM)

in Assay Buffer (50 mM HEPES pH 7.5, 150 mM KGlutamate, 10% glyerol, 1 mM BME). The reactions were incubated for 30 min at 27°C and then centrifuged for 15 min at 18,000 xg to pellet the condensed phase. The supernatant was then removed for analysis and the pellet washed with a 10-fold volumetric excess of Assay Buffer. The centrifugation step was then repeated, the supernatant aspirated, and the pellet resuspended in Protein Buffer (50 mM HEPES pH 7.5, 300 mM KGlutamate, 10% glycerol, 1 mM BME). Finally, the supernatant and/or pelleted material was analyzed by SDS-PAGE analysis and silver stain or blotting (flag-tag on MCM3). Anti-FLAG primary antibody (Sigma-Aldrich F7425) was used at a 1/1,000 dilution and an IRDye800 CW Donkey anti-Rabbit secondary used at 1/10,000. Blots were imaged on a LI-COR Odyssey imager. To assess the contribution of ATP, OCC, or ORC, Cdc6, and Cdt1 separately, the individual reagents were removed from the loading reactions and the experiment completed as described. Blots were quantitated using FIJI (*Schindelin et al., 2012*).

## *Drosophila* genetics

The full genomic coding region of *D. melanogaster* Orc1 including 200 basepairs of downstream regulatory sequence and approximately 1000 basepairs of upstream regulatory sequence was amplified by PCR from OregonR genomic DNA and cloned into the pattB vector with an N-terminal eGFP tag (eGFP-Orc1). To generate eGFP-Orc1$^{WalkerAB}$, the same procedure was used but PCR was utilized to incorporate two point mutations: K604A (Walker A) and D684A/E685A (Walker B); PCR was similarly used to generate eGFP-Orc1$^{\Delta IDR}$, which contains a deletion of the Orc1 N-terminal disordered domain (deletion of residues 249–548). Transgenes were generated by injection (GenetiVision Corporation) for site specific PhiC31 integration at location P23L68A4 on the third chromosome. Homozygous lines were generated to confirm viability and were then balanced and crossed to flies carrying the Orc1$^{4739}$ null allele (*Park and Asano, 2008*) to determine each transgene's capacity to rescue lethality in flies homozygous for Orc1$^{4739}$.

For imaging of embryos by lattice light-sheet microscopy it was desirable to have not only Orc1 tagged with eGFP, but to also be able to visualize histones. We therefore used a recombination cross between flies homozygous for eGFP-Orc1 or eGFP-Orc1$^{WalkerAB}$ and flies containing a His2A-RFP transgene, also residing on the second chromosome. Recombinants containing both eGFP-Orc1/eGFP-Orc1$^{WalkerAB}$ and His2A-RFP were identified by eye color.

## Lattice light-sheet imaging of *D. melanogaster* embryos

Embryos were collected from collection cages after a 90 min laying period. Embryos were dechorionated in 100% bleach for 90 s and placed on the surface of a 5 mm diameter glass coverslip using a fine haired paintbrush. The coverslip surface was made adhesive by applying a small drop of home-made glue solution (prepared by dissolving a roll of double-sided scotch tape in heptane overnight) (*Mir et al., 2018*).

Imaging was performed using a custom built Lattice Light-Sheet Microscope as previously described (*Chen et al., 2014*; *Mir et al., 2018*; *Mir et al., 2017*). For all experiments a 40 beam square lattice pattern was used with a minimum and maximum excitation numerical aperture of 0.44 and 0.55 respectively. The lattice pattern was dithered in the x direction over a 5 μm range in 0.1 μm steps over the duration of each exposure to generate a uniform excitation pattern. Z-stacks were acquired by synchronously scanning the excitation sheet using a galvo-mirror, and the detection objective using a piezo stage over a 25 μm range in 0.250 μm increments. A 488 nm laser was used to excite eGFP (eGFP-Orc1 or eGFP-Orc1$^{WalkerAB}$) with an exposure time of 50 ms at each slice and a 561 nm laser was to excite RFP (His2A-RFP) with an exposure time of 10 ms at each slice. During Z-stack acquisition an image was acquired for both channels at each z-position before moving to the next. The time interval between each stack was 5.67 s. The excitation laser powers were set to 375 μW and 335 μW for the 488 nm and 561 nm lines respectively as measured at the back focal plane of the excitation objective. Images were recorded using two Hamamatsu ORCA-Flash 4.0 digital CMOS cameras. A long-pass dichroic (Semrock FF-560) nm was placed between the two cameras to separate emission wavelengths of above and below 560 nm, and a bandpass emission filter was placed in front of each camera (Semrock FF01-525/50 for eGFP and Semrock FF01-593/46 for RFP).

Once an embryo of suitable age was found (typically starting in nuclear cycle 12 as determined by examining the His2A-RFP channel) it was continuously imaged as described above until the completion of the mitosis preceding nuclear cycle 14.

Images of mitotic events, from metaphase to the completion of telophase were analyzed by first segmenting out chromatin using the His2A-RFP channel (*Figure 5—video 3*). Segmentation was performed using a custom script written in MATLAB R2017B (*Mir, 2019*; copy archived at https://github.com/elifesciences-publications/Parker_2019_MitosisAnalysis). Briefly, first a 3D median filter was applied, followed by 3D Gaussian filtering to each z-stack. A locally adaptive threshold was then calculated, which was then used to generate a binary mask. The binary mask was then filtered by removing regions connected to a border so that only completely imaged nuclei were analyzed, and then filtered for size to remove segmentation errors. A label matrix was then generated from the binary mask for each time point. An implementation of the Hungarian algorithm for tracking (Simple Tracker - File Exchange - MATL...) was then applied to the centroid positions of the individual nuclei in the label matrices to track them through time. The mean intensity values for each nuclei through time were then calculated. Fold enrichment over the background value was calculated for each channel. We found that the local maxima in the eGFP-ORC1 channel intensity corresponded exactly to early telophase in each nucleus (*Figure 5*) and this peak was then used to align each individual trace for averaging. Data from all cleavage events for each line were averaged together as no difference was found between traces from the 13th and 14th divisions in either the eGFP-ORC1 or eGFP-Orc1$^{WalkerAB}$ lines (*Figure 5—figure supplement 1*). A total of 88 and 100 nuclei were analyzed for the eGFP-Orc1 and eGFP-Orc1$^{WalkerAB}$ mutants respectively.

## Imaging Orc1 cellular dynamics in *Drosophila* S2 cells

*Drosophila* S2 cells were maintained as adherent cells at 27°C in ESF 921 medium supplemented to 1% FBS and 1x penicillin-streptomycin at a passage number less than twenty. For transfections, S2 cells were seeded in 6-well dishes at a density of $3 \times 10^6$ cells/well. The medium was replaced 24 hr later and the cells were transfected (Effectene, Qiagen) with two plasmids, one expressing mCherry-tagged *Dm*Histone2A (pCopia_mCherry-H2A was a gift from the lab of Dr. Gary Karpen, UC Berkeley) and another expressing eGFP-tagged Orc1 (pCopia_eGFP-Orc1) (400 ng of each plasmid was used for transfections). pCopia_eGFP-Orc1 was generated by separately PCR amplifying eGFP and Orc1, and using Gibson Assembly to clone into a ligation-independent cloning (LIC) site incorporated into the parental pCopia vector (pCopia-LIC). pCopia_eGFP-Orc1 was then used as a template for around-the-horn mutagenesis to generate Orc1 deletions in the BAH domain (*Orc1$^{\Delta BAH}$*, deletion of amino acids 1–186) and the disordered domain (*Orc1$^{\Delta IDR}$*, deletion of amino acids 248–549). Two sequential rounds of around-the-horn mutagenesis was used to generate an Orc1 construct with mutations in the Walker A (K604A) and Walker B (D684A/E685A) motifs (*Orc1$^{WalkerAB}$*). The transfection mixture was removed 16 hr post-transfection and replaced with fresh medium. After an additional 24 hr incubation, the cells were prepared for imaging by gently resuspending the cells and transferring them to a 35 mm Nunc glass bottom tissue culture dish (ThermoFisher) where they were allowed to adhere for 20 min. After this time, the medium was gently aspirated to remove suspended cells and replaced with fresh ESF 921.

Image stacks were collected on a Zeiss LSM 710 inverted confocal microscope using a 40x oil immersion objective (NA = 1.4) and filter sets for mCherry and eGFP. Samples were scanned at low magnification (20x) using mCherry-H2A signal to identify mitotic cells (<1% of cell population). When metaphase cells were identified they were positioned at the center of the field of view and magnification increased (40x objective with 3x digital zoom). To assess Orc1 dynamics a z-stack (1 μm spaced images over the entire cell volume) was collected every 1.5 min from metaphase through cytokinesis. Images were processed using FIJI. First, each stack of slices was summed and a gaussian blur (radius = 1) filter applied. Subsequently, the average background intensity was subtracted from each channel and an ROI generated for each time point by auto-thresholding the mCherry-H2A channel. The eGFP-Orc1 intensity was measured for each ROI and the fold-change in intensity relative to metaphase intensity was calculated for each time point. We report the average and standard deviation of the maximum fold-change in Orc1 chromosome intensity (observed in telophase) for at least three individual mitotic cells of each transfected construct.

## Phosphorylation assays

The effect of initiator phosphorylation on phase separation was assessed by both the depletion assay and fluorescence microscopy. For depletion assay analysis, phosphorylation reactions were prepared by mixing 1 µM of either ORC, Cdc6, or Cdt1 with 200 nM CDK1/CycA in the presence and absence of 5 mM MgOAc and 1 mM ATP in Protein Buffer (50 mM HEPES pH 7.5, 300 mM KGlutamate, 10% glycerol, 1 mM BME). The phosphorylation reactions were incubated for 60 mins at room temperature and then, to set up the depletion assay, the reactions were mixed with dsDNA and incubated for 30 min at room temperature. The final depletion assay reaction conditions include 500 nM initiator, 125 nM CDK1/CycA, 2.5 mM MgOAc, 500 nM ATP, and 500 nM dsDNA in Assay Buffer (50 mM HEPES pH 7.5, 300 mM KGlutamate, 10% glycerol, 1 mM BME). After incubation, the samples were processed as previously described, assaying protein depletion by SDS-PAGE analysis and Silver staining. For assaying the effect of phosphorylation by fluorescence microscopy, protein concentrations were scaled up 4-fold for the phosphorylation reaction (4 µM ORC and 0.8 µM CDK1/CycA); all other reaction conditions are as described. Subsequently, the phosphorylation reactions were diluted 1:1 with equimolar Cy5-dsDNA in Assay Buffer, incubated for 2 min, and 7 µL were spotted on a glass slide, covered with a glass coverslip, and imaged as described previously.

## Acknowledgements

We thank past and present members of the Berger and Botchan labs for helpful discussion and advice. We also thank Alison Killilea of the UC Berkeley Cell Culture facility for assistance with insect cell cultures. Parent pCopia and pCopia_mCherry-H2A vectors for work in *Drosophila* S2 tissue culture were a gift from the lab of Dr. Gary Karpen, UC Berkeley. Eric Greene (lab of Andy Martin, UC Berkeley) assisted in fluorescent anisotropy DNA-binding measurements. We thank Steve Bell for providing purified yeast ORC for analysis of phase separation behavior. This work was supported by an NIH NRSA postdoctoral fellowship (F32GM116393, to MWP) and by the NCI (R01CA030490, to JMB and MRB).

## Additional information

### Competing interests

James M Berger: Reviewing editor, *eLife*. Michael R Botchan: Reviewing editor, *eLife*. The other authors declare that no competing interests exist.

### Funding

| Funder | Grant reference number | Author |
|---|---|---|
| National Institute of General Medical Sciences | F32GM116393 | Matthew W Parker |
| National Cancer Institute | R01CA030490 | Michael R Botchan James M Berger |

The funders had no role in study design, data collection and interpretation, or the decision to submit the work for publication.

### Author contributions

Matthew W Parker, Conceptualization, Formal analysis, Funding acquisition, Validation, Investigation, Visualization, Methodology, Writing—original draft, Writing—review and editing; Maren Bell, Jonchee A Kao, Formal analysis, Investigation; Mustafa Mir, Software, Formal analysis, Investigation; Xavier Darzacq, Supervision; Michael R Botchan, Conceptualization, Supervision, Funding acquisition, Methodology, Writing—original draft, Writing—review and editing; James M Berger, Conceptualization, Supervision, Funding acquisition, Methodology, Writing—original draft, Project administration, Writing—review and editing

## Author ORCIDs

Matthew W Parker (iD) https://orcid.org/0000-0002-7571-0010
Mustafa Mir (iD) http://orcid.org/0000-0001-8280-2821
Xavier Darzacq (iD) http://orcid.org/0000-0003-2537-8395
Michael R Botchan (iD) https://orcid.org/0000-0003-0459-5518
James M Berger (iD) https://orcid.org/0000-0003-0666-1240

## Decision letter and Author response

Decision letter https://doi.org/10.7554/eLife.48562.023
Author response https://doi.org/10.7554/eLife.48562.024

## Additional files

### Supplementary files

• Transparent reporting form
DOI: https://doi.org/10.7554/eLife.48562.021

### Data availability

All data generated during this study is included in the manuscript and supporting files.

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
