## [Decision Letter]

Thank you for submitting your article "A new class of disordered elements controls DNA replication through initiator self-assembly" for consideration by *eLife*. Your article has been reviewed by three peer reviewers, including Stephen Bell as the Reviewing Editor and Reviewer #1, and the evaluation has been overseen by Jessica Tyler as the Senior Editor.

The reviewers have discussed the reviews with one another and the Reviewing Editor has drafted this decision to help you prepare a revised submission.

Summary:

The manuscript by Parker et al. convincingly demonstrates that the proteins required for metazoan origin licensing can phase separate in vitro in a DNA-dependent manner and when present together they phase separate into the same droplets. Interestingly, the Mcm2-7 helicase that drives all DNA unwinding at eukaryotic replication forks can only partition into the so formed droplets if all three of the other proteins required for origin licensing are present. This implies that droplet formation is important for Mcm2-7 recruitment. In addition, the authors observe Orc1 foci in vivo which depends on the protein's disordered region as does phase separation in vitro, thus connecting their biochemical studies with physiology. The authors also identify intrinsic-disordered regions required for this phase separation and find that they have a shared composition. Interestingly, the authors find that cyclin-dependent kinase activation negatively regulates phase separation in vitro Even though the in vivo data is sparse, the in vitro experiments carried out are very convincing and the manuscript is well written. Taken together, the authors shows, for the first time, that phase separation plays a role in DNA replication which can explain some previously made observations. This constitutes a new aspect of DNA replication and integrates well with current knowledge.

Essential revisions:

1) Throughout the in vitro experiments, a 60 bp dsDNA is used. The authors should explain why they picked this sequence specifically and include additional experiments showing whether the sequence or length of the added DNA impacts phase separation by ORC, Cdc6 or Cdt1. Whether sequence is important or not, the authors should discuss the implications of these findings with regard to their model.

2) The authors claim that Orc1 association with chormatin was fully abolished when its IDR is deleted. However, the images shown suggest a residual, albeit weaker, binding during telophase. Which images have been used to generate Figure 5I? Anaphase images? Is the result the same when telophase images are used?

3) In the Discussion, the authors state "The present work establishes […] that DNA replication […] is regulated by macromolecular phase separation." While the authors show that ORC localization is altered by the IDR, while likely, more experiments would be necessary to demonstrate that DNA replication as a whole is regulated. The authors should tone this statement down or provide more evidence to support this conclusion.

Other points:

1) The authors provide an appropriately nuanced discussion of the meaning of chromatin association of ORC versus the DNA binding required for helicase loading. One suggestion to improve this discussion (and that will be interesting to explore experimentally in the future) is the role of naked versus nucleosomal DNA in driving these phase transitions. Metazoan ORC is thought to localize to nucleosome-depleted DNA and, if naked DNA versus nucleosomal DNA has different abilities to drive phase transitions, it seems possible that the authors' observations might help to explain a preference for non-nucleosomal DNA for ORC DNA binding.

2) Could these same LLPS possibly be used for the entire DNA replication process? Or do the authors have reason to believe that they exist only in G1, or only for the origin licensing process?

3) The Mcm10 protein appears to travel with forks, and although not noted to this reviewer's knowledge, it appears to have what appears to be long disordered regions that border both sides the central globular DNA binding region. This factor is essential to replication from yeast to human. Perhaps this is one factor that may participate in LLPS, possibly with this same LLPS for PreRC formation, enabling replication foci often visualized by PCNA staining in cells. Perhaps the authors would like to comment on possible extension of the PreRC LLPS to possible replication within this particle?

4) Phosphorylation inhibits phase separation of ORC especially. Yeast also control ORC function by CDK phosphorylation, but one might assume this is a distinct process since phase separation doesn't apparently occur in budding yeast. A brief comment on this different mechanism but related use of CDK to inhibit ORC function would be helpful for general readership.

5) Possibly too speculative – but could there be a correlation between the facts that Sc ORC1 does not phase separate, but that S.c. is unique in ability to recognize specific origin sequences, while higher organisms might utilize phase separation elements instead?

6) In Figure 1C & D, the authors measure the K_d_ of Cdt1 binding to DNA. However, in their gel shift assay they clearly measure how much DNA is necessary to induces phase separation of Cdt1 rather than the affinity of Cdt1 to DNA. This is likely true for the fluorescence polarisation assay as well (which the authors incorrectly call fluorescence anisotropy in the text). The authors should clearly state / discuss this point to avoid confusion. In addition, it would be interesting to see if Cdt1 can also interact with DNA outside the context of phase separation and how strong this interaction is.

7) Figure 2E & G, 3E, 4F & G and others: the authors should help the readers understanding the figures by putting more labels. E.g. in Figure 2E, write DNA next to - +. And in Figure 4F, write something like α-FLAG next to the Western blot.

8) Figure 2F: the x-axis label is misleading, especially when compared to Figure 2G. Not only is the Cdt1 concentration varied but the DNA concentration and DNA are also present. The reader has to pay a lot of attention to the text to understand this.

9) The authors mention the use of DISOPRED too late in the text. It should be mentioned when first used (Figure 1A).

10) Figure 3A: The authors use% of disorder in the whole protein chain. However, this may not be the most useful metric and total IDR length is more meaningful. Later in the manuscript, the authors switch to discussing this instead of percentage. Figure 3A should thus be changed accordingly.

11) Figure 3—figure supplement 1F: Is the reduction of DNA binding in the presence of ATP really significant?

12) Figure 3B: label the lanes as this is used to discuss the figure in the text.

13) Figure 3G and H: mention the protein concentrations in the figures.

14) Figure 4A: The labelling of this figure is difficult to understand. The authors should mention the proteins used (Cdc6 and Cdt1) and make clear that the first column contains protein and DNA but not Cdc6 and Cdt1. As currently labeled the panel is misleading.

15) Consider moving Figure 4E and 6E to the supplements and Supplementary Figure 5G into the main figures. The finding that FUS does not go into the droplets is quite significant as it suggests specificity of protein recruitment. This should be emphasised more in the discussion.

16) Figure 5A: The text mentions the use of RFP-H2A but the figure is labelled with mCherry-H2A. Which fluorophore was used?

17) The authors should describe in the materials and methods how the intracellular Cdt1 concentration shown in Figure 2—figure supplement 1H was determined.

18) Figure 6D: The lanes corresponding to lanes in Figure 6C should be labeled with a-d to help the reader understand the figure.

19) Supplementary Figure 5G: The order of images is inconsistent with the rest of the manuscript (imaging channels left to right instead of top to bottom) and thus confusing.

---

## [Author Response]

Essential revisions:1) Throughout the in vitro experiments, a 60 bp dsDNA is used. The authors should explain why they picked this sequence specifically and include additional experiments showing whether the sequence or length of the added DNA impacts phase separation by ORC, Cdc6 or Cdt1. Whether sequence is important or not, the authors should discuss the implications of these findings with regard to their model.

We thank the reviewers for bringing up this point. Since metazoan origins and initiator DNA-binding profiles are not sequence specific, we began our studies with a random dsDNA oligonucleotide. We chose 60 bp as a starting point because prior studies have shown that this length is modestly greater than is required for the ATP-dependent binding of *Drosophila* ORC (Bleichert et al., 2018) and for OCCM formation (e.g., Yuan et al., 2017). We have now clarified in the main text that this sequence was randomly generated and have included additional experiments examining the effect of both DNA sequence and length on initiator phase separation as the reviewers request (Figure 1—figure supplement 1 and Figure 3—figure supplement 1). We find that a variety of DNA sequences can induce ORC, Cdc6, and Cdt1 phase separation to an equal extent. Additionally, we show that optimal phase separation of ORC and Cdt1 requires dsDNA oligonucleotides longer than 25 bp. By comparison, Cdc6 is more sensitive to DNA length, showing weak phase separation with oligonucleotides 25-60 base pairs in length, but more robust LPC formation in the presence of plasmid DNA. These new data are now discussed within the context of our model for initiator DNA/chromatin-binding and with respect to what is known regarding ORC’s preference for nucleosome free chromatin regions.

2) The authors claim that Orc1 association with chormatin was fully abolished when its IDR is deleted. However, the images shown suggest a residual, albeit weaker, binding during telophase. Which images have been used to generate Figure 5I? Anaphase images? Is the result the same when telophase images are used?

We agree that there does appear to be a residual amount of ORC^ΔOrc1IDR^ present on chromosomes. However, the quantitation shown in Figure 5I reports on the telophase levels of Orc1, when maximum loading is observed in both S2 cells and embryos. To further clarify ORC dynamics at different stages in mitosis, and to aid in the interpretation of these data, we have now included quantitative time course analysis of the recruitment of ORC to chromosomes in mitotic S2 cells (Figure 5—figure supplement 1A). These data reveal that although ORC, ORC^ΔBAH^, and ORC^WalkerAB^ are loaded onto chromosomes beginning in anaphase, the levels of ORC^ΔOrc1IDR^ actually decrease in anaphase and then recover in telophase to the level observed in metaphase. This recovery to background levels accounts for the residual binding that is seen in the images. In addition to these data, we have also included a better explanation of what and how quantitation was done in both the figure legend and Materials and methods.

3) In the Discussion, the authors state "The present work establishes […] that DNA replication […] is regulated by macromolecular phase separation." While the authors show that ORC localization is altered by the IDR, while likely, more experiments would be necessary to demonstrate that DNA replication as a whole is regulated. The authors should tone this statement down or provide more evidence to support this conclusion.

We agree that this statement is a bit overly strong, and that additional experimentation is needed to unequivocally demonstrate a role of phase separation in metazoan DNA replication. We have revised the sentence to read:

“The present work provides strong evidence that DNA replication, a cellular pathway vital to cell proliferation, is impacted by phase separation-promoting elements.”

Other points:1) The authors provide an appropriately nuanced discussion of the meaning of chromatin association of ORC versus the DNA binding required for helicase loading. One suggestion to improve this discussion (and that will be interesting to explore experimentally in the future) is the role of naked versus nucleosomal DNA in driving these phase transitions. Metazoan ORC is thought to localize to nucleosome-depleted DNA and, if naked DNA versus nucleosomal DNA has different abilities to drive phase transitions, it seems possible that the authors' observations might help to explain a preference for non-nucleosomal DNA for ORC DNA binding.

We agree with that this is a very interesting question of future inquiry. We have now substantially expanded our discussion of ORC origin selection to speculate on how our observations of ORC phase separation on naked DNA might relate to the in vivo preference of ORC for nucleosome free regions, and to point out that it will be necessary to conduct future studies of LPC formation in the context of single nucleosomes and chromatin.

2) Could these same LLPS possibly be used for the entire DNA replication process? Or do the authors have reason to believe that they exist only in G1, or only for the origin licensing process?

We again agree that this is a very interesting issue. We initially refrained from commenting on the point in our original draft, as we did not want to be overly speculative. However, it has not escaped our notice that many proteins required for metazoan DNA replication have intrinsically disordered regions (including, as noted in the next point, Mcm10). Given the selectivity we observe for initiator condensates, we think it plausible that the phase formed by the Pre-RC machinery may represent an evolving structure that, through the fluid exchange of its resident components, drives the replication reaction and culminates in the spatial coordination of multiple active replication forks that could serve as replication factories. We have taken the reviewers’ suggestion/comment as a license for speculation and have now added the following paragraph in the Discussion:

“From a parallel perspective, it is interesting to note that the visualization of both newly-replicated DNA (Kennedy et al., 2000; Xiang et al., 2018) and proteins present at the replication fork during S-phase (Chagin et al., 2016) often reveals a non-uniform organization within the nucleus, a feature that has led to the proposal that replisomes may be grouped into ‘factories’ (Cook, 1999). […] Given the selectivity we observe for initiator condensates (Figure 4), it is possible that the phase formed by Pre-RC machineries may represent an evolving structure that, through the fluid exchange of its resident components facilitated by IDR-IDR interactions, drives replication forward, culminating in the spatial co-localization of multiple active forks within a nuclear zone.”3) The Mcm10 protein appears to travel with forks, and although not noted to this reviewer's knowledge, it appears to have what appears to be long disordered regions that border both sides the central globular DNA binding region. This factor is essential to replication from yeast to human. Perhaps this is one factor that may participate in LLPS, possibly with this same LLPS for PreRC formation, enabling replication foci often visualized by PCNA staining in cells. Perhaps the authors would like to comment on possible extension of the PreRC LLPS to possible replication within this particle?

We agree – please see above response to the point above.

4) Phosphorylation inhibits phase separation of ORC especially. Yeast also control ORC function by CDK phosphorylation, but one might assume this is a distinct process since phase separation doesn't apparently occur in budding yeast. A brief comment on this different mechanism but related use of CDK to inhibit ORC function would be helpful for general readership.This is a very pertinent point – thank you for bringing it up. The following comment has been added in the Discussion to clarify how the differing abilities of yeast and metazoan ORC to phase separate might influence our understanding of phospho-regulation mechanisms: “Budding yeast ORC is also negatively regulated by CDK-dependent phosphorylation (Nguyen et al., 2001), although this regulation likely occurs through a distinct mechanism, as we did not observe phase separation of ScORC (Figure 3—figure supplement 1E).”5) Possibly too speculative – but could there be a correlation between the facts that Sc ORC1 does not phase separate, but that S.c. is unique in ability to recognize specific origin sequences, while higher organisms might utilize phase separation elements instead?The reviewers’ comment zeroes in on a fundamental question: what is the biological rationale for evolving the use of phase separation in DNA replication? We agree that one possible explanation is that as multi-cellularity evolved, LPC formation might serve to offset a loss of specific origin sequences, which could be cumbersome to cells with large genomes and more complex and malleable transcriptional profiles. However, we confess that we haven’t found a reasonable way to discuss the issue that doesn’t build on a house of cards. A key line of inquiry that will be important to answering the “why” of initiator phase separation will be to establish when in evolution phase separation appeared – i.e., is it a metazoan-specific phenomenon, or can some “simpler” eukaryotic species (such as *S. pombe*) also utilize such a mechanism? For now, we would prefer not to delve into this point in the present manuscript.6) In Figure 1C & D, the authors measure the K_d_ of Cdt1 binding to DNA. However, in their gel shift assay they clearly measure how much DNA is necessary to induces phase separation of Cdt1 rather than the affinity of Cdt1 to DNA. This is likely true for the fluorescence polarisation assay as well (which the authors incorrectly call fluorescence anisotropy in the text). The authors should clearly state / discuss this point to avoid confusion. In addition, it would be interesting to see if Cdt1 can also interact with DNA outside the context of phase separation and how strong this interaction is.

The reviewers bring up an interesting experimental issue: to what extent are classic binding protein-DNA equilibria altered by phase separation? These two phenomena could be partly disentangled by using separation of function mutants. We currently lack point mutants for the proteins under investigation here that might be used in such a capacity (one such set might be phospho-mimetic mutants in CDK sites within the IDRs). We might also try to conduct EMSA studies in the presence of hexanediol, which is commonly used to disrupt LPC formation. At the present, systematically conducting such efforts is beyond the scope of the present work. We do now include an additional competition experiment that tests the equilibrium of the system and hence whether the K_d_’s we determined are reasonable approximations (Figure 1—figure supplement 1D). These results suggest that equilibrium is maintained and that the calculated K_d_’s are reasonable starting approximations. Nonetheless, we have relabeled all K_d_’s as apparent K_d_’s or “K_d_, app” to reflect the ambiguity of this point.

As a final response to this point, we edited the main text to correctly read “fluorescence polarization”, not “anisotropy”.

7) Figure 2E & G, 3E, 4F & G and others: the authors should help the readers understanding the figures by putting more labels. E.g. in Figure 2E, write DNA next to - +. And in Figure 4F, write something like α-FLAG next to the Western blot.

Thank you. Additional annotation has been added to these figures.

8) Figure 2F: the x-axis label is misleading, especially when compared to Figure 2G. Not only is the Cdt1 concentration varied but the DNA concentration and DNA are also present. The reader has to pay a lot of attention to the text to understand this.

The x-axis label has now been edited to reflect that the concentrations of both Cdt1 and DNA are titrated.

9) The authors mention the use of DISOPRED too late in the text. It should be mentioned when first used (Figure 1A).DISOPRED is now mentioned in the main text for Figure 1A.10) Figure 3A: The authors use% of disorder in the whole protein chain. However, this may not be the most useful metric and total IDR length is more meaningful. Later in the manuscript, the authors switch to discussing this instead of percentage. Figure 3A should thus be changed accordingly.We agree with the reviewers that the length of the longest continuous disordered domain in a protein is possibly more important than percent disorder overall. We have therefore calculated the longest region of disorder for each component of the Pre-RC and have included this information in Figure 3A, as well as a discussion of these data in the main text.11) Figure 3—figure supplement 1F: Is the reduction of DNA binding in the presence of ATP really significant?We did not intend to indicate that there is a significant difference for DNA binding in the presence of ATP and we apologize for the confusion. To clarify this point, we have now added to the main text: “Similar to budding yeast Cdc6 (Feng et al., 2000), DmCdc6 was found to interact with DNA by dsDNA-coupled agarose bead pull-down assays; the presence of ATP had no significant effect on Cdc6 DNA-binding (Figure 3—figure supplement 1F).”12) Figure 3B: label the lanes as this is used to discuss the figure in the text.Done.13) Figure 3G and H: mention the protein concentrations in the figures.Done.14) Figure 4A: The labelling of this figure is difficult to understand. The authors should mention the proteins used (Cdc6 and Cdt1) and make clear that the first column contains protein and DNA but not Cdc6 and Cdt1. As currently labeled the panel is misleading.We thank the reviewers for this excellent suggestion. We have now revised and expanded the annotation to Figure 4A (and Figure 4B) as recommended. This markedly improves the readability of the figure.15) Consider moving Figure 4E and 6E to the supplements and Supplementary Figure 5G into the main figures. The finding that FUS does not go into the droplets is quite significant as it suggests specificity of protein recruitment. This should be emphasised more in the discussion.

We have now moved the specificity data of Supplemental Figure 5G to Figure 4. Bringing these data to the forefront of the manuscript represents a significant improvement and we thank the reviewers for this comment. We do, however, find that the Cdc6 phosphorylation result, while inconclusive, completes the data presented in Figure 6. We have therefore opted to retain this panel with the main figures.

16) Figure 5A: The text mentions the use of RFP-H2A but the figure is labelled with mCherry-H2A. Which fluorophore was used?

We thank the reviewers for pointing out this incongruency. We were in fact using an RFP-tagged H2A line. The figure has been changed accordingly.

17) The authors should describe in the materials and methods how the intracellular Cdt1 concentration shown in Figure 2—figure supplement 1H was determined.

A new section has now been added to the Materials and methods explaining our method in detail. We now state:

“Determination of embryonic Cdt1 concentration. *D. melanogaster* embryos (Bloomington stock 32045) were collected at 2 hr intervals from age 2-16 hrs, dechorionated in 100% bleach for 60 sec, and washed extensively with H_2_O. […] Blots were imaged with a LI-COR Odyssey imager. A standard curve was generated from the intensity values of the recombinant Cdt1 dilution series and the per- embryo Cdt1 concentration calculated based on the reported volume of a *D. melanogaster* embryo (Markow et al., 2009).”

18) Figure 6D: The lanes corresponding to lanes in Figure 6C should be labeled with a-d to help the reader understand the figure.

Done.

19) Supplementary Figure 5G: The order of images is inconsistent with the rest of the manuscript (imaging channels left to right instead of top to bottom) and thus confusing.

Done. This panel has now been moved to the main text and the images reorganized as suggested.